# Disentangling Shared Representations Improves Implicit Neural Representations for Medical Imaging

## Abstract

Implicit neural representations (INRs) have emerged as a powerful paradigm for medical imaging via physics-informed unsupervised learning. Classical INRs optimize an entire network from scratch for each subject, leading to inefficient training and suboptimal imaging quality. Recent initialization-based approaches attempt to inject population priors into pre-trained networks, yet they rely on high-quality images and often suffer from catastrophic forgetting during fine-tuning. We present DisINR, a novel INR framework that explicitly disentangles shared and subject-specific representations. DisINR introduces a shared encoder–decoder pair and subject-specific encoders, whose features are jointly decoded for image reconstruction. By integrating differentiable forward models, it pre-trains the shared modules directly from limited raw measurements, removing the need for pre-acquired high-quality images. During test-time adaptation, only the subject-specific encoder is optimized, while the shared pair remains frozen, effectively preserving learned priors. Extensive evaluations on three representative medical imaging tasks show that DisINR significantly outperforms state-of-the-art INRs in both reconstruction accuracy and efficiency.

## 1. Introduction

Medical imaging is a cornerstone of modern clinical practice, enabling the detailed visualization of internal anatomical structures of the human body (Harisinghani et al., 2019; Rubin, 2014). Inverse problems in medical imaging aim to recover anatomical images from raw measurements (*e.g.*, , projections in CT or $k$-space data in MRI). However, due

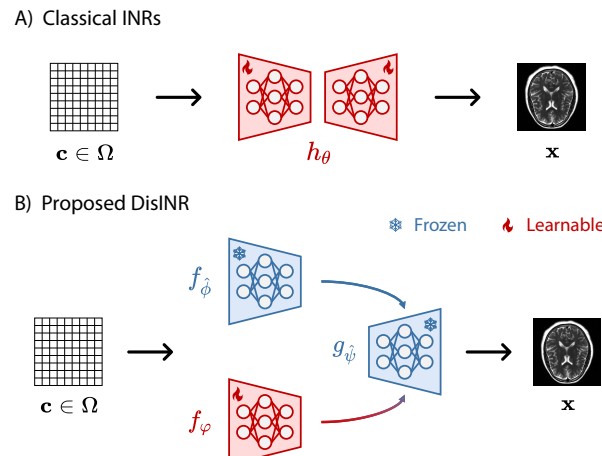

*Figure 1.* **A)** Classical INRs optimize an entire network $h_\theta$ from scratch for each subject; **B)** The proposed DisINR inherits a pre-trained encoder–decoder pair ($f_{\hat{\phi}}, g_{\hat{\psi}}$) shared across diverse subjects, while learning only a subject-specific encoder $f_\varphi$ for the target, effectively leveraging population priors and thus producing improved reconstructions.

to factors such as undersampling, these problems are inherently ill-posed with multiple suboptimal solutions (Huang et al., 2024; Wang et al., 2023), calling for more advanced computational reconstruction algorithms.

Recently, implicit neural representations (INRs) have shown great potential in solving medical inverse problems through physics-informed unsupervised learning (Molaei et al., 2023; Luo et al., 2025). INRs model each image as a continuous function parameterized by a coordinate-based neural network. With differentiable forward models (*e.g.*, , Fourier transform for MRI), the network can be optimized without using external training data. Owing to the intrinsic bias of neural networks toward structured image patterns (Rahaman et al., 2019; Mildenhall et al., 2021), INRs can resolve high-quality images in an unsupervised way. Yet, existing INR approaches (Zha et al., 2022; Feng et al., 2023; Cai et al., 2024) typically optimize an entire network from scratch for each subject, which leads to low efficiency and limited reconstruction quality.

A promising direction is to learn an effective network ini-

[1]Anonymous Institution, Anonymous City, Anonymous Region, Anonymous Country. Correspondence to: Anonymous Author <anon.email@domain.com>.

Preliminary work. Under review by the International Conference on Machine Learning (ICML). Do not distribute.

tialization from pre-acquired high-quality images. Early studies (Tancik et al., 2021; Lee et al., 2021; Chen & Wang, 2022) employ meta-learning, which performs well but requires a large number of pre-training images. More recently, Vyas et al. (2024) proposed STRAINER, which learns transferable representations via a shared encoder and subject-specific decoders. This approach enables INRs to achieve a strong initialization even from just a few images. However, both types of initialization techniques face two key challenges: **1) Dependence on high-quality images.** Such diagnosis-quality images are often difficult to obtain in medical scenarios, particularly for rare diseases; **2) Catastrophic forgetting.** The population priors encoded in the initialized network are easily overwritten during case-specific fine-tuning, as all network parameters are updated at test time. These limitations reduce practical applicability and potential performance improvements.

In this work, we propose DisINR, a novel INR framework that explicitly disentangles shared and subject-specific representations. Specifically, DisINR introduces a shared encoder–decoder pair along with subject-specific encoders, whose features are jointly decoded to reconstruct images. By integrating differentiable forward models, DisINR can pre-train the encoder-decoder pair shared across diverse subjects directly from a limited set of raw measurements. This can effectively alleviate the reliance on high-quality images, significantly improving the model's applicability in a wide range of clinical scenarios. During test-time adaptation, we freeze the pre-trained pair, while optimizing only a subject-specific encoder for a given unseen test sample. This design fundamentally eliminates the catastrophic forgetting problem inherent in existing initialization techniques (Tancik et al., 2021; Vyas et al., 2024). Therefore, our method can effectively incorporate learned population priors into INR optimization, enabling faster and higher-quality reconstructions. We evaluate DisINR on three representative medical imaging tasks, including 3D volume fitting, undersampled MRI, and sparse-view CT. Extensive experiments demonstrate that DisINR substantially outperforms state-of-the-art INRs in both reconstruction accuracy and efficiency.

## 2. Related Work

### 2.1. Medical Inverse Problems

The data acquisition process of a medical imaging system can generally be formulated as below:

$$\mathbf{y} = \boldsymbol{A}\mathbf{x} + \boldsymbol{\epsilon}, \tag{1}$$

where $\mathbf{y} \in \mathbb{R}^m$ denotes measured data, $\mathbf{x} \in \mathbb{R}^n$ is the desired image, $\boldsymbol{A} \in \mathbb{R}^{m \times n}$ is the system matrix (i.e., , forward model), and $\boldsymbol{\epsilon} \in \mathbb{R}^m$ represents system noise. Note that the formulations are presented in the real domain for simplicity, but they also apply to complex-valued data, such

as $k$-space measurements in MRI.

Medical inverse problems aim to reconstruct unknown images $\mathbf{x}$ from measurements $\mathbf{y}$. Due to various factors like undersampling (i.e., , $m \ll n$), these inverse problems are often highly ill-posed, where multiple suboptimal solutions exist. Conventional model-based algorithms (Beister et al., 2012; Fessler, 2010; Thibault et al., 2007) formulate it as the following optimization problem:

$$\hat{\mathbf{x}} = \arg\min_{\mathbf{x}} \left[ \|\boldsymbol{A}\mathbf{x} - \mathbf{y}\|_1 + \lambda \cdot \mathcal{R}(\mathbf{x}) \right], \tag{2}$$

where $\hat{\mathbf{x}}$ denotes the optimal solution, $\mathcal{R}$ is an explicit regularizer (e.g., , total variation (Rudin et al., 1992) for image smoothness), and $\lambda$ is a hyperparameter controlling the contribution of the regularizer. The use of the regularizer can effectively constrain the solution space, thus enabling improved image reconstructions. However, such handcrafted regularizers often fail to capture the complex distribution of medical images, thereby limiting the resulting image quality.

### 2.2. INR for Medical Inverse Problems

As a signal representation way based on neural networks, implicit neural representations (INRs) have shown great potential in solving medical inverse problems, such as undersampled MRI (Shen et al., 2022; Huang et al., 2023; Feng et al., 2023; Liu et al., 2025; Feng et al., 2025; Wu et al., 2025) and CT (Sun et al., 2021; Zang et al., 2021; Shen et al., 2022; Zha et al., 2022; Wu et al., 2023; Cai et al., 2024; Du et al., 2024). Technically, INR models an image as a continuous function of spatial coordinates: $\mathbf{c} \in \Omega \to \mathbf{x}$, and uses a coordinate-based neural network $\boldsymbol{h}_\theta$ to learn this mapping. Formally, INR solves the optimization problem:

$$\hat{\theta} = \arg\min_{\theta} \|\boldsymbol{A}\boldsymbol{h}_\theta(\mathbf{c}) - \mathbf{y}\|_1, \tag{3}$$

where $\hat{\theta}$ denotes the optimized network parameters. Under this formulation, recovering the image $\mathbf{x}$ is transformed into optimizing the parameters $\theta$. Therefore, the desired image can be resolved as $\hat{\mathbf{x}} = \boldsymbol{h}_{\hat{\theta}}(\mathbf{c})$ in an unsupervised manner. The key insight is that neural networks possess an inherent and generalizable inductive bias toward continuous image structures (Rahaman et al., 2019; Mildenhall et al., 2021), which serves as an implicit prior enabling high-quality reconstructions.

### 2.3. Advances for INR Initialization

As shown in Eq. (3), classical INRs for medical reconstruction optimize an independent network from random Gaussian initialization, i.e., , $\theta \sim \mathcal{N}(0, 1)$, for each subject. To enhance reconstruction efficiency and accuracy, recent works explore INR initialization strategies that generally fall

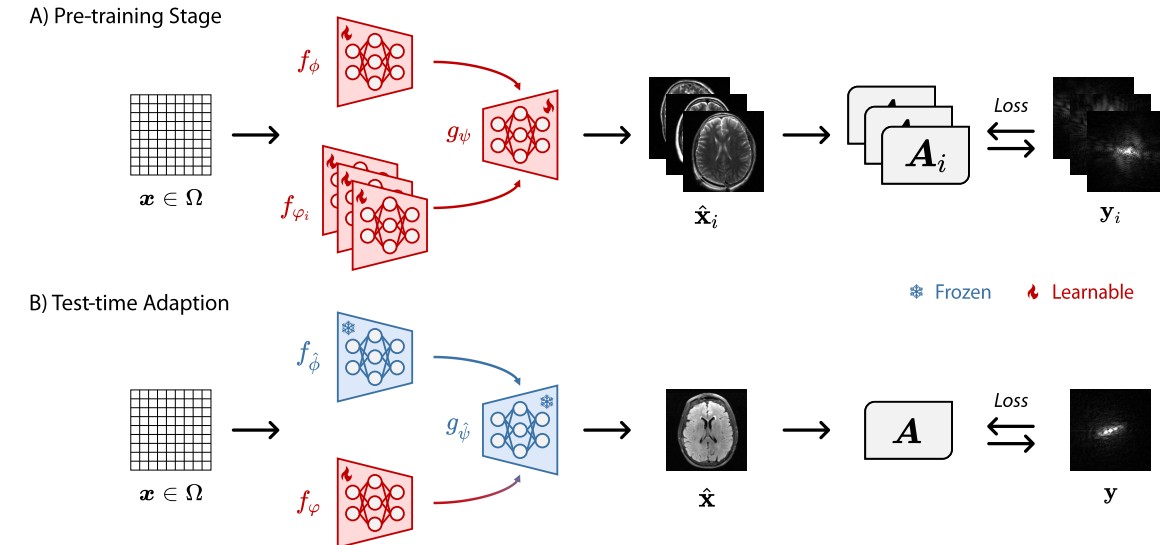

*Figure 2.* Overview of the proposed DisINR, which consists of a shared encoder–decoder pair $(\boldsymbol{f}_\phi, \boldsymbol{g}_\psi)$ and multiple subject-specific encoders $\{\boldsymbol{f}_{\varphi_i}\}_{i=1}^N$. **A) Pre-training Stage:** All modules in DisINR are jointly optimized from a few raw measurements $\mathbf{Y} = \{\mathbf{y}_i\}_{i=1}^N$ by incorporating differentiable forward models $\{\boldsymbol{A}_i\}_{i=1}^N$. **B) Test-time Adaption:** Given a new measurement $\mathbf{y} \notin \mathbf{Y}$ and the corresponding forward model $\boldsymbol{A}$, the shared pair $(\boldsymbol{f}_{\hat\phi}, \boldsymbol{g}_{\hat\psi})$ is frozen while only the subject-specific encoder $\boldsymbol{f}_\varphi$ is optimized. This effectively injects learned population priors into the INR optimization process, enabling faster and higher-quality reconstruction $\hat{\mathbf{x}}$.

into two categories: 1) Meta-learning (Tancik et al., 2021; Lee et al., 2021; Chen & Wang, 2022; Kim et al., 2023). For example, Tancik et al. (2021) showed that MAML- and Reptile-based meta-learning can significantly accelerate INR optimization and boost reconstruction quality across diverse tasks; 2) Transerable feature learning (Vyas et al., 2024; 2025; Rangarajan et al., 2025). STRAINER (Vyas et al., 2024) adopts a shared encoder and individual decoders, enabling feature transfer from a few high-quality images. However, these initialization methods depend on diagnosis-quality images and tend to overfit (*i.e.*, , catastrophic forgetting) during test-time adaptation, severely limiting their applicability and performance.

## 3. Method

### 3.1. Overview

Given a set $\mathbf{Y} = \{\mathbf{y}_i\}_{i=1}^N$ consisting of a limited number of undersampled measurements (*e.g.*, , projections in CT or $k$-space data in MRI), our goal is to reconstruct the corresponding high-quality image $\mathbf{x}$ from a new measurement $\mathbf{y} \notin \mathbf{Y}$ in a fully unsupervised way. This inverse problem involves two key challenges: **1)** How to extract rich population priors directly from the set $\mathbf{Y}$? **2)** How to inject the learned population priors into the model optimization?

To achieve this, we propose DisINR, a new INR framework that explicitly disentangles shared and subject-specific representations. Unlike conventional INR methods (Feng

et al., 2023; Cai et al., 2024) for medical reconstructions, which optimize an independent network $\boldsymbol{h}_\theta$ from scratch for each subject, as shown in Fig. 1, our DisINR inherits a pre-trained encoder–decoder pair $(\boldsymbol{f}_{\hat\phi}, \boldsymbol{g}_{\hat\psi})$, while learning only a subject-specific encoder $\boldsymbol{f}_\varphi$ for the target. Formally, given a target $\mathbf{x}$, DisINR represents it as follows:

$$\mathbf{x} = \boldsymbol{g}_{\hat\psi}(\boldsymbol{f}_{\hat\phi}(\mathbf{c}) \odot \boldsymbol{f}_\varphi(\mathbf{c})), \tag{4}$$

where $\odot$ denotes the concatenation operator, $\hat\phi$ and $\hat\psi$ represent the parameters of the pre-trained pair shared across diverse subjects, and $\varphi$ are the parameters of the encoder specialized for the target $\mathbf{x}$.

By explicitly disentangling shared and subject-specific representations, our framework enables flexible transfer of the shared representation provided by the pre-trained encoder–decoder pair into subject-specific optimization during test-time adaptation. This design thus enables faster and improved reconstructions.

### 3.2. Pre-taining Stage

Suppose the measurement set $\mathbf{Y}$ and the corresponding forward model set $\boldsymbol{A} = \{\boldsymbol{A}_i\}_{i=1}^N$, we first seek to embed the population priors inherent in the set $\mathbf{Y}$ into the shared encoder-decoder pair of DisINR. Fig. 2**A** illustrates this pre-training stage.

For each sample $\mathbf{y}_i \in \mathbf{Y}$, we first assign a specific encoder $\boldsymbol{f}_{\varphi_i}$. Both the shared encoder $\boldsymbol{f}_\phi$ and the subject-specific

**Algorithm 1 Pre-training Stage**

**Input:** Raw measurements $\mathbf{Y} = \{\mathbf{y}_i\}_{i=1}^N$, forward models $\{\mathbf{A}_i\}_{i=1}^N$, coordinates $\mathbf{c} \in \Omega$, and learning rate $\beta$

**Output:** Learned parameters $\hat{\phi}, \hat{\psi}$, and $\{\hat{\varphi}_i\}_{i=1}^N$

1: Initialize $f_\phi$, $g_\psi$, and $\{f_{\varphi_i}\}_{i=1}^N$ with random Gaussian-distributed parameters, *i.e.*, , $\phi, \psi, \varphi_i \sim \mathcal{N}(0,1), \forall i$.
2: **for** iteration $t = 1$ to $T$ **do**
3:    **for** subject $i = 1$ to $N$ **do**
4:       $\hat{\mathbf{x}}_i \leftarrow g_\psi(f_\phi(\mathbf{c}) \odot f_{\varphi_i}(\mathbf{c}))$
5:       $\mathcal{L}_i \leftarrow \|\mathbf{A}_i\hat{\mathbf{x}}_i - \mathbf{y}_i\|_1$
6:    **end for**
7:    $\varphi_i \leftarrow \varphi_i - \beta\nabla_{\varphi_i}\mathcal{L}_i, \ \forall i$
8:    $\phi, \psi \leftarrow \phi, \psi - \beta\nabla_{\phi,\psi}\sum_{i=1}^N \mathcal{L}_i$
9: **end for**
10: **return** $\hat{\phi}, \hat{\psi}, \{\hat{\varphi}_i\}_{i=1}^N$

**Algorithm 2 Test-time Adaptation**

**Input:** New measurement $\mathbf{y} \notin \mathbf{Y}$, forward model $\mathbf{A}$, coordinates $\mathbf{c} \in \Omega$, learning rate $\beta$, and frozen shared encoder-decoder pair $(f_{\hat{\phi}}, g_{\hat{\psi}})$,

**Output:** Learned parameters $\hat{\varphi}$ and image $\hat{\mathbf{x}}$

1: Initialize $f_\varphi$ with random Gaussian-distributed parameters, *i.e.*, , $\varphi \sim \mathcal{N}(0,1)$.
2: **for** iteration $t = 1$ to $T$ **do**
3:    $\varphi \leftarrow \varphi - \beta\nabla_\varphi\|\mathbf{A}g_{\hat{\psi}}(f_{\hat{\phi}}(\mathbf{c}) \odot f_\varphi(\mathbf{c})) - \mathbf{y}\|_1$
4: **end for**
5: **return** $\hat{\varphi}, \hat{\mathbf{x}} = g_{\hat{\psi}}(f_{\hat{\phi}}(\mathbf{c}) \odot f_{\hat{\varphi}}(\mathbf{c}))$

specific encoder. Hence, the reconstruction is given by

$$\hat{\mathbf{x}} = \boldsymbol{g}_{\hat{\psi}}(\boldsymbol{f}_{\hat{\phi}}(\mathbf{c}) \odot \boldsymbol{f}_{\hat{\varphi}}(\mathbf{c})). \qquad (7)$$

The procedure of the test-time adaptation is presented in Algorithm 2. By freezing the shared encoder-decoder pair, we can fundamentally mitigate the problem of catastrophic forgetting commonly observed in initialization-based techniques (Tancik et al., 2021; Vyas et al., 2024). This strategy effectively incorporates population priors into INR optimization, thereby enhancing reconstruction quality and efficiency.

### 3.4. Network Architecture

The proposed DisINR framework is **architecture-agnostic**, allowing it to be seamlessly integrated with different INR backbones. Owing to its efficient multi-resolution hash encoding, neural graphics primitive (NGP) (Müller et al., 2022) has currently become one of the leading INR architectures in various medical imaging tasks, such as sparse-view CT (Zha et al., 2022; Cai et al., 2024; Wu et al., 2023) and undersampled MRI (Feng et al., 2023; 2025).

In this study, we thus use NGP as the backbone of DisINR. Both the shared encoder $\boldsymbol{f}_\phi$ and the subject-specific encoder $\boldsymbol{f}_{\varphi_i}$ share the same design, consisting of a multi-resolution hash encoding module followed by a two-layer MLP. The shared decoder $\boldsymbol{g}_\psi$ is also implemented as a two-layer MLP, which takes as input the concatenated features from the two encoders. Besides, *all network configurations are kept consistent across experiments*, demonstrating the generalization and robustness of DisINR. Additional implementation details are provided in the Appendix.

## 4. Experiments

In this section, we evaluate the effectiveness and generalization of DisINR on three representative medical imaging tasks: 3D volume fitting (Sec. §4.1), undersampled MRI (Sec. §4.2), and sparse-view CT (Sec. §4.3). We also analyze the effects of network architecture on model perfor-

encoder $\boldsymbol{f}_{\varphi_i}$ take the spatial coordinates at imaging space $\mathbf{c} \in \Omega$ as inputs, generating shared and subject-specific representations $\boldsymbol{f}_\phi(\mathbf{c})$ and $\boldsymbol{f}_{\varphi_i}(\mathbf{c})$, respectively. The decoder $\boldsymbol{g}_\psi$ then maps the concatenated features to the reconstructed image as $\hat{\mathbf{x}}_i = \boldsymbol{g}_\psi(\boldsymbol{f}_\phi(\mathbf{c}) \odot \boldsymbol{f}_{\varphi_i}(\mathbf{c}))$. Next, the predicted image $\hat{\mathbf{x}}_i$ is projected to the measurement domain through the differentiable forward operator $\boldsymbol{A}_i$. Finally, we jointly optimize the three networks by minimizing the prediction error in the measurement domain. Formally, this process can be expressed as follows:

$$\hat{\psi}, \hat{\phi}, \{\hat{\varphi}_i\}_{i=1}^N = \underset{\phi,\psi,\{\varphi_i\}_{i=1}^N}{\arg\min} \left[\sum_{i=1}^N \|\boldsymbol{A}_i\hat{\mathbf{x}}_i - \mathbf{y}_i\|_1\right], \quad (5)$$
$$\hat{\mathbf{x}}_i = \boldsymbol{g}_\psi(\boldsymbol{f}_\phi(\mathbf{c}) \odot \boldsymbol{f}_{\varphi_i}(\mathbf{c})),$$

where $\hat{\phi}$, $\hat{\psi}$, and $\{\hat{\varphi}_i\}_{i=1}^N$ denote the learned parameters of the shared pair and the subject-specific encoders, respectively. The detailed procedure of the pre-training stage is presented in Algorithm 1.

### 3.3. Test-time Adaptation

After the model pre-training, the shared encoder $\boldsymbol{f}_{\hat{\phi}}$ and decoder $\boldsymbol{g}_{\hat{\psi}}$ capture rich representations that generalize across diverse subjects. During test-time adaptation, our goal is to recover the corresponding high-quality image $\hat{\mathbf{x}}$ from a new measurement $\mathbf{y} \notin \mathbf{Y}$, without using any external data.

As shown in Fig. 2B, we freeze the shared encoder-decoder pair $(\boldsymbol{f}_{\hat{\phi}}, \boldsymbol{g}_{\hat{\psi}})$, and optimize only a new subject-specific encoder $\boldsymbol{f}_\varphi$ from scratch, *i.e.*, , $\varphi \sim \mathcal{N}(0,1)$, by incorprating the differentiable forwad model $\boldsymbol{A}$. Mathematically, we solve the following optimization problem:

$$\hat{\varphi} = \underset{\varphi}{\arg\min} \|\boldsymbol{A}\boldsymbol{g}_{\hat{\psi}}(\boldsymbol{f}_{\hat{\phi}}(\mathbf{c}) \odot \boldsymbol{f}_\varphi(\mathbf{c})) - \mathbf{y}\|_1, \quad (6)$$

where $\hat{\varphi}$ represents the learned parameters of the subject-

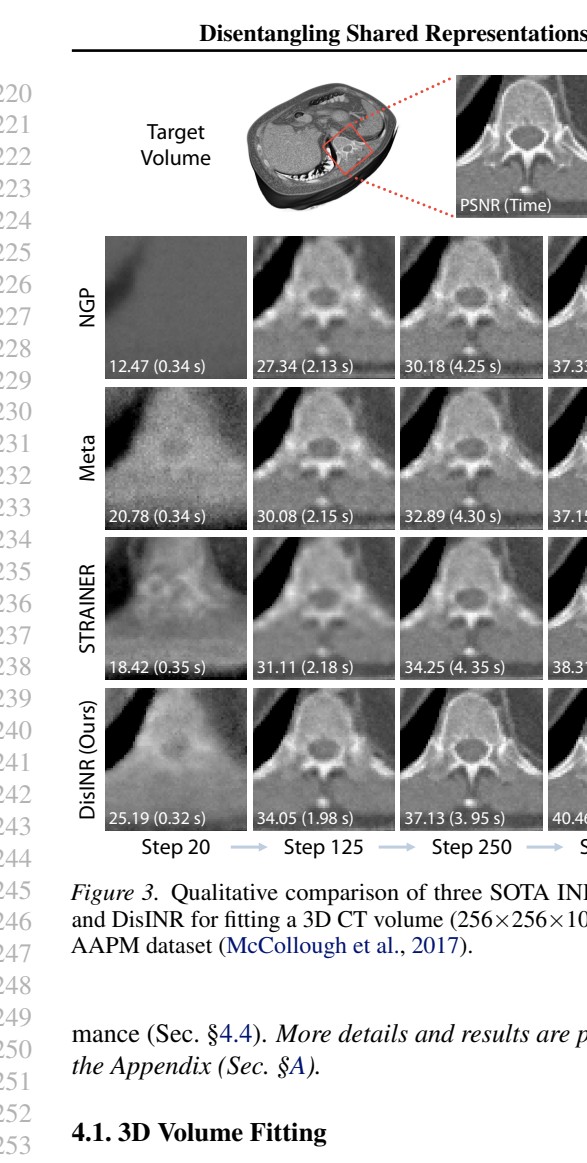

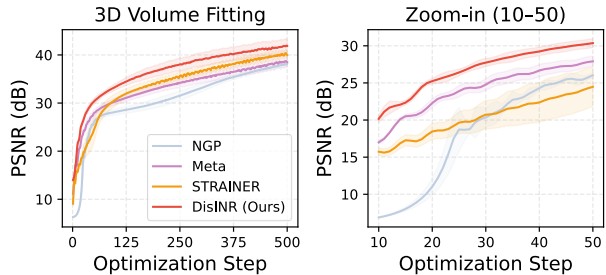

*Figure 4.* Performance curves of three SOTA INR baselines and DisINR for fitting 3D CT volumes (256×256×100) from the AAPM dataset (McCollough et al., 2017).

*Table 1.* Comparison of learnable parameters and running time between three SOTA INR baselines and DisINR during the pre-training and test stages for 3D volume fitting on the AAPM dataset (McCollough et al., 2017).

| Method | Pretraining | | Test | |
|---|---|---|---|---|
| | # Param. | Time | # Param. | Time |
| NGP | – | – | 23.40 M | 8.5 s |
| Meta | 23.40 M | 558.3 s | 23.40 M | 8.6 s |
| STRAINER | 23.63 M | **124.9** s | 23.63 M | 8.7 s |
| DisINR (Ours) | **21.71 M** | 186.5 s | **10.83 M** | **7.9** s |

*Figure 3.* Qualitative comparison of three SOTA INR baselines and DisINR for fitting a 3D CT volume (256×256×100) from the AAPM dataset (McCollough et al., 2017).

mance (Sec. §4.4). *More details and results are provided in the Appendix (Sec. §A).*

### 4.1. 3D Volume Fitting

Unlike natural 2D RGB images, medical images are typically represented as 3D volumetric data. The increased dimensionality poses a significant challenge for achieving efficient representations. Here, we evaluate the performance of DisINR on the 3D volume fitting task.

**Datasets** We employ 9 human body CT volumes from the AAPM dataset (McCollough et al., 2017), each with a size of 256×256×100. The volumes are first thresholded using a Hounsfield Unit (HU) window of [-800, 400] and then normalized to the range [0, 1]. Finally, we split these volumes into two subsets: 6 volumes for pre-training and 3 volumes for test.

**Compared Methods** we use three cutting-edge INR methods as baselines: 1) NGP (Müller et al., 2022), a powerful grid-based INR architecture; 2) Meta (Tancik et al., 2021), an INR initialization framework based on meta-learning; and 3) STRAINER (Vyas et al., 2024), an SOTA INR model for learning transferable representations. Note that both the

original Meta and STRAINER are implemented with SIREN backbones (Sitzmann et al., 2020). For a fair comparison, we re-implement them using the NGP backbone (Müller et al., 2022).

**Results** Fig. 3 provides the qualitative comparisons. Visually, DisINR reconstructs clear anatomical structures with sharp tissue boundaries as early as step 125, whereas other methods still produce blurry images. Fig. 4 shows the optimization curves of all methods on the 3D volume fitting task. DisINR converges much faster, reaching over 30 dB in PSNR within 50 steps, while others remain around 25 dB. After full convergence (500 steps), it finally outperforms the second-best method, STRAINER (Vyas et al., 2024), by more than 2 dB. This indicated that the shared representations learned by DisINR provide a strong prior, helping the model converge faster and achieve higher reconstruction accuracy.

Also, as summarized in Table 1, DisINR also exhibits clear advantages in computational efficiency. At the pre-training stage, it maintains a compact model size (21.71M parameters) with moderate training time (186.5 s). During test-time adaptation, only the subject-specific encoder is fine-tuned, reducing the number of learnable parameters to 10.83M and achieving the fastest inference speed (7.9 s). In contrast, other methods require updating the entire network, leading to higher memory usage and slower inference.

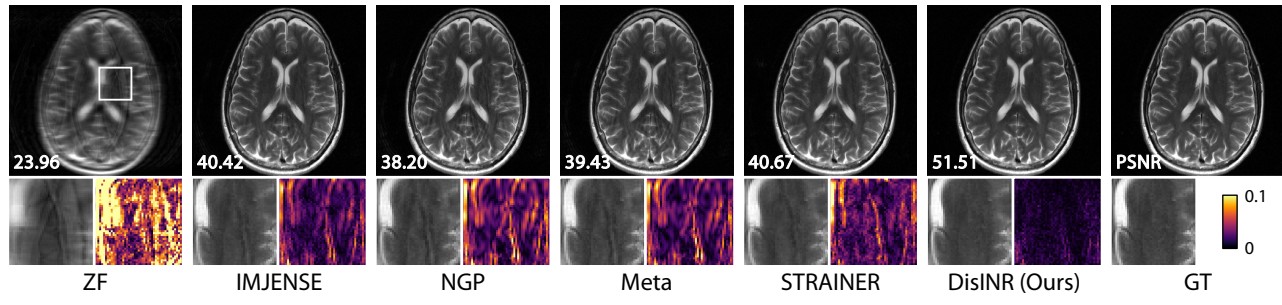

*Figure 5.* Quantitative comparison of five baselines (including analytical ZF and four SOTA INR models) and DisINR for undersampled MRI with a Cartesian pattern of AF = 6× on a representative sample of the fastMRI-T2w dataset (Knoll et al., 2020).

*Table 2.* Quantitative comparison of five baselines (including analytical ZF and four SOTA INR models) and DisINR for undersampled MRI on the fastMRI-T2w and fastMRI-FLAIR datasets (Knoll et al., 2020).

| Method | fastMRI-T2w (in-domain) | | | | fastMRI-FLAIR (out-of-domain) | | | |
|---|---|---|---|---|---|---|---|---|
| | AF = 6× | | AF = 8× | | AF = 6× | | AF = 8× | |
| | PSNR | SSIM | PSNR | SSIM | PSNR | SSIM | PSNR | SSIM |
| ZF | $23.47_{\pm1.51}$ | $0.681_{\pm0.036}$ | $23.12_{\pm1.43}$ | $0.660_{\pm0.036}$ | $21.24_{\pm3.57}$ | $0.684_{\pm0.045}$ | $21.06_{\pm3.54}$ | $0.655_{\pm0.048}$ |
| IMJENSE | $38.76_{\pm5.28}$ | $0.961_{\pm0.041}$ | $29.30_{\pm2.84}$ | $0.848_{\pm0.049}$ | $41.61_{\pm6.72}$ | $0.979_{\pm0.022}$ | $29.29_{\pm5.04}$ | $0.901_{\pm0.055}$ |
| NGP | $36.95_{\pm4.74}$ | $0.947_{\pm0.043}$ | $28.38_{\pm2.49}$ | $0.818_{\pm0.047}$ | $41.36_{\pm7.59}$ | $0.977_{\pm0.024}$ | $28.87_{\pm5.10}$ | $0.891_{\pm0.059}$ |
| MetaNGP | $37.42_{\pm4.78}$ | $0.951_{\pm0.042}$ | $28.56_{\pm2.57}$ | $0.824_{\pm0.049}$ | $42.20_{\pm8.06}$ | $0.980_{\pm0.023}$ | $29.48_{\pm5.46}$ | $0.900_{\pm0.059}$ |
| STRAINER | $40.22_{\pm3.78}$ | $0.972_{\pm0.032}$ | $31.36_{\pm2.99}$ | $0.892_{\pm0.041}$ | $37.28_{\pm5.17}$ | $0.969_{\pm0.015}$ | $28.17_{\pm4.90}$ | $0.886_{\pm0.046}$ |
| DisINR (Ours) | $\mathbf{48.42}_{\pm5.18}$ | $\mathbf{0.992}_{\pm0.021}$ | $\mathbf{35.46}_{\pm3.96}$ | $\mathbf{0.952}_{\pm0.036}$ | $\mathbf{45.64}_{\pm4.79}$ | $\mathbf{0.990}_{\pm0.009}$ | $\mathbf{33.55}_{\pm4.82}$ | $\mathbf{0.946}_{\pm0.026}$ |

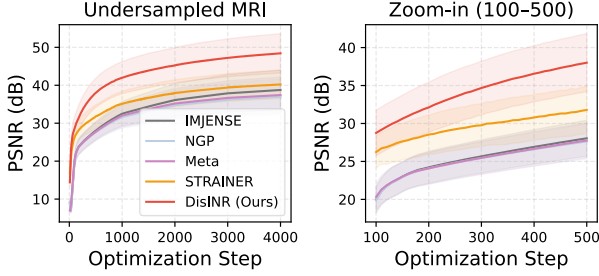

*Figure 6.* Performance curves of four SOTA INR baselines and DisINR for undersampled MRI with a Cartesian of AF = 6× on the fastMRI-T2w dataset (Knoll et al., 2020).

## 4.2. Undersampled MRI

MRI acquisition is inherently slow due to physical constraints. Undersampled MRI can accelerate the scanning process, but reconstructing high-quality MR images from incomplete *k*-space data remains a challenging ill-posed inverse problem. Here, we evaluate the performance of DisINR on the undersampled MRI reconstruction.

**Datasets** The fastMRI dataset (Knoll et al., 2020) is one of the most commonly used public datasets for MRI reconstruction, containing large-scale multi-contrast *k*-space data. Here, we use 150 multi-coil 2D brain *k*-space samples of

size 256×256 from the fastMRI dataset (Knoll et al., 2020), including 100 T2w and 50 FLAIR scans. We simulate a 1D uniform Cartesian pattern with acceleration factors (AF) of 6 and 8. The dataset is divided into three subsets: 50 T2w samples for pre-training, 50 T2w samples for in-domain test, and 50 FLAIR samples for out-of-domain test.

**Compared Methods** In addition to the three INR methods (*i.e.*, , NGP (Müller et al., 2022), Meta (Tancik et al., 2021), and STRAINER (Vyas et al., 2024)) used in the 3D volume fitting task, we also compare two baselines designed for MRI reconstruction: 1) Zero-Filling (ZF), a classical MRI alrgothrm; and 2) IMJENSE (Feng et al., 2023), a SOTA INR-based model for parallel MRI. The original IMJENSE is built upon the SIREN architecture (Sitzmann et al., 2020). We reproduce IMJENSE using the NGP backbone (Müller et al., 2022) for a fair comparison.

**Results** Table 2 reports the quantitative comparisons. Overall, DisINR achieves the best performance across all AF and domains. On the in-domain fastMRI-T2w dataset, DisINR attains 48.42 dB at AF = 6 and 35.46 dB at AF = 8, outperforming the second-best method, STRAINER (Vyas et al., 2024), by more than 8 dB. On the out-of-domain fastMRI-FLAIR dataset, where the image contrast differs from the pre-training T2w data, DisINR still demonstrates excellent generalization. It achieves 45.64 dB at AF = 6 and

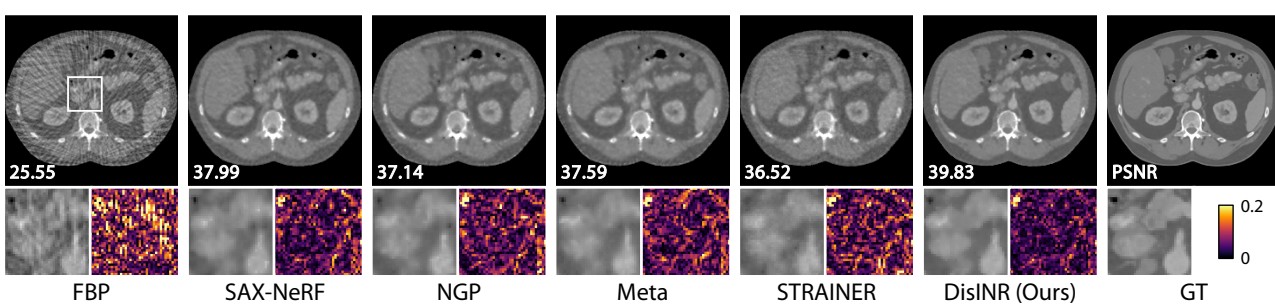

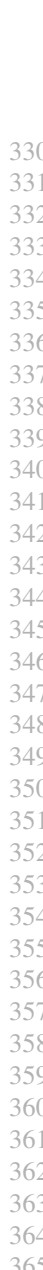

*Figure 7.* Qualitative comparison of five baselines (including analytical FBP and four SOTA INR models) and DisINR for sparse-view CT with 60 projection views on a representative sample of the DeepLesion dataset (Yan et al., 2018).

*Table 3.* Quantitative comparison of five baselines (including analytical FBP and four SOTA INR models) and DisINR for sparse-view CT on the DeepLesion (Yan et al., 2018) and LIDC (Armato III et al., 2011) datasets.

| Method | DeepLesion (in-domain) | | | | LIDC (out-of-domain) | | | |
|---|---|---|---|---|---|---|---|---|
| | 60 Views | | 90 Views | | 60 Views | | 90 Views | |
| | PSNR | SSIM | PSNR | SSIM | PSNR | SSIM | PSNR | SSIM |
| FBP | $23.98_{\pm1.25}$ | $0.362_{\pm0.039}$ | $26.70_{\pm1.25}$ | $0.478_{\pm0.043}$ | $24.47_{\pm2.03}$ | $0.403_{\pm0.062}$ | $27.40_{\pm2.39}$ | $0.524_{\pm0.076}$ |
| SAX-NeRF | $37.71_{\pm1.60}$ | $0.965_{\pm0.010}$ | $38.92_{\pm1.49}$ | $0.975_{\pm0.007}$ | $37.34_{\pm3.00}$ | $0.948_{\pm0.036}$ | $38.13_{\pm3.02}$ | $0.956_{\pm0.033}$ |
| NGP | $36.89_{\pm1.42}$ | $0.957_{\pm0.010}$ | $38.09_{\pm1.34}$ | $0.969_{\pm0.007}$ | $36.50_{\pm2.73}$ | $0.940_{\pm0.036}$ | $37.50_{\pm2.74}$ | $0.951_{\pm0.033}$ |
| Meta | $36.96_{\pm1.35}$ | $0.958_{\pm0.010}$ | $38.08_{\pm1.34}$ | $0.969_{\pm0.007}$ | $36.61_{\pm2.72}$ | $0.941_{\pm0.036}$ | $37.52_{\pm2.69}$ | $0.951_{\pm0.032}$ |
| STRAINER | $36.17_{\pm1.46}$ | $0.945_{\pm0.014}$ | $37.05_{\pm1.40}$ | $0.956_{\pm0.011}$ | $35.35_{\pm2.77}$ | $0.920_{\pm0.043}$ | $36.17_{\pm2.76}$ | $0.932_{\pm0.039}$ |
| DisINR (Ours) | $\mathbf{39.24}_{\pm1.65}$ | $\mathbf{0.976}_{\pm0.008}$ | $\mathbf{40.30}_{\pm1.54}$ | $\mathbf{0.982}_{\pm0.006}$ | $\mathbf{38.02}_{\pm2.97}$ | $\mathbf{0.954}_{\pm0.034}$ | $\mathbf{38.85}_{\pm2.98}$ | $\mathbf{0.962}_{\pm0.031}$ |

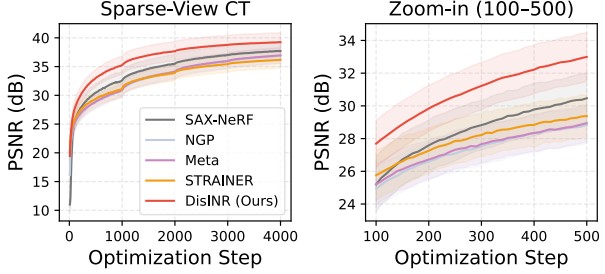

*Figure 8.* Performance curves of four SOTA INR baselines and DisINR for sparse-view CT with 60 projection views on the DeepLesion dataset (Yan et al., 2018).

33.55 dB at AF = 8, exceeding all other methods by 3~5 dB. These results confirm that DisINR delivers higher accuracy and stronger generalization to unseen domains.

Fig. 6 shows the PSNR evolution curves of all methods over optimization steps on the in-domain fastMRI-T2w dataset at AF = 6×. DisINR not only converges significantly faster but also achieves the highest final reconstruction accuracy among all compared methods. The qualitative results are presented in Fig. 5. Visually, DisINR effectively removes undersampling artifacts while preserving both global structures and fine anatomical details. Its reconstructed images are visually closest to the GTs.

### 4.3. Sparse-View CT

CT scanning inevitably involves ionizing radiation. Sparse-view sampling effectively reduces the radiation dose, but it often leads to severe streak artifacts in reconstructed images due to undersampling. Here, we evaluate the performance of DisINR on the sparse-view CT reconstruction.

**Datasets** The DeepLesion (Yan et al., 2018) and LIDC (Armato III et al., 2011) datasets are two widely used public CT benchmarks, containing diverse human body scans collected from multiple clinical institutions. Specifically, we extract 100 and 50 slices of size 256×256 from the two datasets, respectively. Each slice is thresholded using a Hounsfield Unit (HU) window of [-800, 400] and normalized to the range [0, 1]. To generate raw projections, we simulate a 2D fan-beam geometry with 60 and 90 views using the CIL toolbox (Jørgensen et al., 2021; Papoutsellis et al., 2021). The dataset is divided into three subsets: 50 cases from DeepLesion for pre-training, 50 cases from DeepLesion for in-domain test, and 50 cases from LIDC for out-of-domain test.

**Compared Methods** In addition to the three INR methods (*i.e.*, , NGP (Müller et al., 2022), Meta (Tancik et al., 2021), and STRAINER (Vyas et al., 2024)) used in the 3D volume fitting task, we also compare two baselines designed for CT

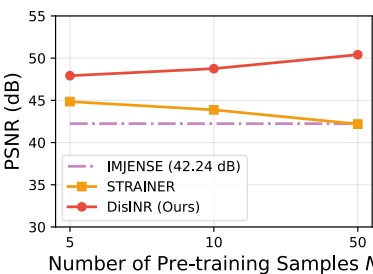

*Figure 9.* Performance curves of IMJENSE (Feng et al., 2023), STRAINER (Vyas et al., 2024), and DisINR under different numbers of pre-training samples $N$ for undersampled MRI with a Cartesian pattern of AF = 6× on the fastMRI-T2w dataset (Knoll et al., 2020).

reconstruction: 1) FBP, an analytical CT algorithm; and 2) SAX-NeRF (Cai et al., 2024), an SOTA INR-based method based on a line Transformer. SAX-NeRF is based on the official implementation.

**Results** Quantitative results are shown in Fig. 7 and Table 3. DisINR achieves the best PSNR and SSIM in all settings, clearly outperforming the second-best method while keeping anatomical details sharper and reducing streak artifacts. Visual examples in Fig. 7 also show that DisINR preserves tissue boundaries and fine structures that other methods often oversmooth or distort. In addition, the optimization curves in Fig. 8 indicate that DisINR converges faster and reaches higher accuracy with fewer iterations, showing its stable and data-efficient reconstruction.

### 4.4. Discussion

**Effect of Pre-training Data Size** To study how the size of the pre-training sample affects reconstruction quality, we conduct experiments on the fastMRI-T2w dataset (Knoll et al., 2020) using a Cartesian pattern with AF = 6×. We vary the number of available pre-training subjects ($N$ = 5, 10, 50) and compare our DisINR with IMJENSE (Feng et al., 2023) and STRAINER (Vyas et al., 2024). Note that IMJENSE (Feng et al., 2023) does not involve any pre-training stage and thus serves as a fixed baseline.

Fig. 9 reports the quantitative results. DisINR consistently achieves the highest PSNR under all settings, surpassing IMJENSE (Feng et al., 2023) and STRAINER (Vyas et al., 2024) by a clear margin, and its performance increases monotonically with $N$ from 47.92 dB (5 samples) to 50.41 dB (50 samples). In contrast, STRAINER shows no gain from more pre-training data and even degrades slightly (44.85 to 42.20 dB), indicating the limited scalability of its shared-encoder and individual-decoder design. This confirms DisINR's scalability and its ability to effectively extract and reuse population priors for robust reconstruction.

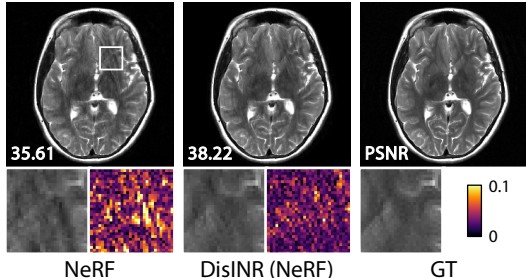

*Figure 10.* Qualitative comparison of NeRF (Mildenhall et al., 2021) and DisINR implemented by the NeRF backbone for undersampled MRI with a Cartesian pattern of AF = 6× on a representative sample of the fastMRI-T2w dataset (Knoll et al., 2020).

*Table 4.* Quantitative comparison of NeRF (Mildenhall et al., 2021) and DisINR implemented by the NeRF backbone for undersampled MRI with a Cartesian pattern of AF = 6× on the fastMRI-T2w dataset (Knoll et al., 2020).

| Method | PSNR | # Param. |
|---|---|---|
| NeRF | $34.97_{\pm 3.11}$ | 0.34 M |
| DisINR (NeRF) | $\mathbf{37.42_{\pm 2.74}}$ | **0.31 M** |

**Effect of Network Backbone** Our DisINR framework is architecture-agnostic, and we validate its effectiveness using different INR backbones. We implement DisINR with the NeRF backbone (Mildenhall et al., 2021), which consists of a positional encoder followed by an MLP: specifically, both encoders use positional encoding followed by a 5-layer MLP, and the decoder is a 2-layer MLP.

As shown in Table 4, DisINR (NeRF backbone) achieves higher PSNR than vanilla 6-layer NeRF (37.42 vs. 34.97 dB) with fewer parameters (0.31M vs. 0.34M). This indicates that population priors can be effectively integrated into different INR architectures. Fig. 10 further shows that DisINR yields clearer anatomical structures and fewer undersampling artifacts, demonstrating its architectural generality.

### 5. Conclusion

In this work, we present DisINR, a new method to incorporate population priors into INR learning for medical imaging. Using physics-informed unsupervised learning, DisINR effectively pre-trains a shared encoder–decoder pair to capture rich representations directly from a limited amount of raw measurements. By freezing the learned encoder–decoder pair, a subject-specific encoder is then optimized for each new subject. Evaluations on three classic medical imaging tasks demonstrate that DisINR achieves SOTA performance compared to existing INR techniques in both accuracy and efficiency.

## Impact Statement

This paper presents work whose goal is to advance the field of Machine Learning. There are many potential societal consequences of our work, none which we feel must be specifically highlighted here.

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

*Figure 11.* 1D Cartesian sampling patterns with acceleration factors of 6 and 8 used in the undersampled MRI task.

# A. Appendix

## A.1. Limitation

A key limitation of this work lies in the limited exploration of DisINR's scalability with respect to the size of the pre-training dataset. Although we conduct preliminary examination of the effect of pre-training data size (from 5 to 50 subjects), comprehensive studies on large-scale datasets remain to be conducted. A more extensive, large-scale validation is left for future work.

## A.2. Data Pre-processing

In our experiments, we include two classical medical imaging tasks: undersampled MRI and sparse-view CT. Here, we detail the data pre-processing. **We will release our code upon acceptance to improve reproducibility**.

**Undersampled MRI**  Given multi-coil 2D brain $k$-space data of size $256\times256$ from the fastMRI dataset (Knoll et al., 2020), as shown in Fig. 11, we simulate 1D Cartesian sampling patterns with acceleration factors of 6 and 8, where the size of the auto-calibration region (ACS) is set to 24. While coil-sensitivity maps computed from the raw fully sampled $k$-space data are used for parallel MRI.

**Sparse-view CT**  For raw 2D slices from the DeepLesion (Yan et al., 2018) and LIDC (Armato III et al., 2011) datasets, we generate sparse-view CT projections by using the CIL toolbox (Jørgensen et al., 2021; Papoutsellis et al., 2021) to simulate a 2D fan-beam geometry. The detailed acquisition parameters are provided in Table 5.

## A.3. Quantitative Metrics

In our evaluation, we use peak signal-to-noise ratio (PSNR) and structural similarity index (SSIM) (Wang et al., 2004), two widely used visual metrics, to quantitatively assess the model performance. Specifically, for undersampled MRI reconstruction, we calculate these two metrics using the normalized amplitude map, as MRI images are complex-valued. For CT reconstruction (including 3D volume fitting and sparse-view CT), we directly compute them using the

*Table 5.* Acquisition parameters of 2D fan-beam geometry used in the sparse-view CT.

| Parameters | Values |
|---|---|
| Type of geometry | 2D fan-beam |
| Image Size | $256\times256$ |
| Voxel Size (mm$^2$) | $1\times1$ |
| View Range ($°$) | $[0, 360)$ |
| Number of Detectors | 500 |
| Detector Spacing (mm) | 2 |
| Number of Projection Views | 60/90 |
| Distance from Source to Center (mm) | 300 |
| Distance from Center to Detector (mm) | 300 |

reconstructed CT images without applying additional pre-processing.

## A.4. Implementations of Proposed DisINR

The proposed DisINR framework is **architecture-agnostic**, allowing it to be seamlessly integrated with different INR backbones. In this study, we adopt NGP (Müller et al., 2022) as the backbone of DisINR. Specifically, both the shared encoder $f_\phi$ and the subject-specific encoder $f_{\varphi_i}$ are composed of a hash encoding module followed by two fully connected (FC) layers, each containing 128 neurons with ReLU activation. The output dimension of each encoder is 128. The shared decoder $g_\psi$ consists of two FC layers: the first layer has 128 hidden units with ReLU activation, and the output layer is linear (without activation). The decoder takes as input the concatenated feature vector from the shared and subject-specific encoders, resulting in an input dimension of 256. For the hash encoding, we use the following configuration: number of levels $L = 10$, number of entries per level $T = 2^{18}$, feature dimension $F = 8$, minimum resolution $N_{\min} = 2$, and per-level factor $b = 2$.

For model optimization, we employ the Adam optimizer (Kingma, 2014) with its default hyperparameters. The learning rate is initialized to $1 \times 10^{-3}$ and decayed by a factor of 0.5 every 1,000 iterations. The network is trained for a total of 4,000 iterations. *Note that the same optimization configurations are used for both the pre-training and test-time adaptation stages. All configurations are kept consistent across all experiments*, further demonstrating the generalization and robustness of DisINR.

## A.5. Additional Visual Results

Fig. 13 and Fig. 12 show additional qualitative comparisons between our DisINR and the baselines for the undersampled MRI and sparse-view CT tasks. Visually, our DisINR produces the best reconstructions, which are closest to the GT samples.

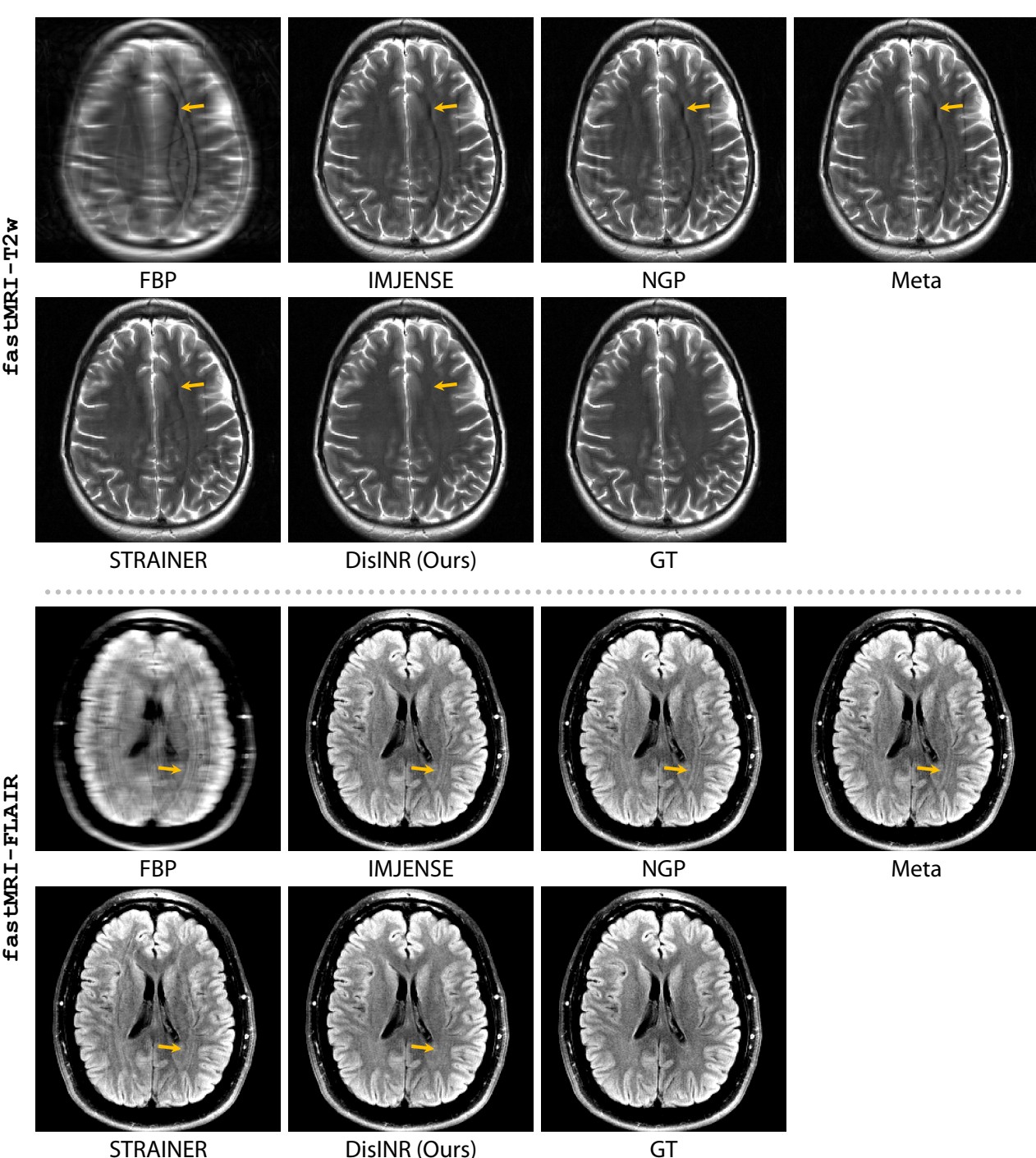

*Figure 12.* Quantitative comparison of five baselines (including analytical ZF and four SOTA INR models) and our DisINR for undersampled MRI with a Cartesian pattern of AF = 6× on two representative samples of the fastMRI-T2w and fastMRI-FLAIR datasets (Knoll et al., 2020).

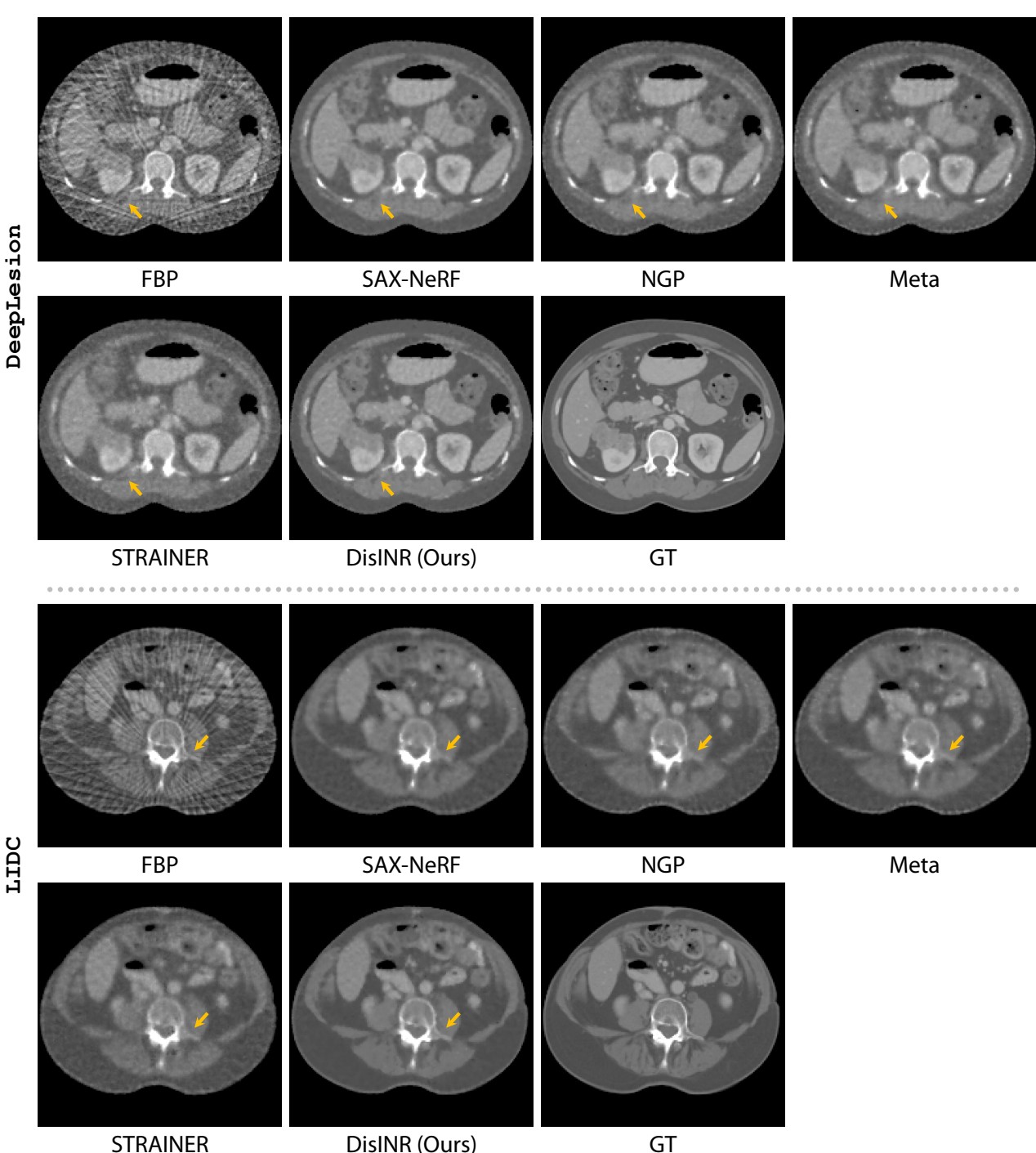

*Figure 13.* Qualitative comparison of five baselines (including analytical FBP and four SOTA INR models) and our DisINR for sparse-view CT with 60 projection views on two representative samples of the DeepLesion (Yan et al., 2018) and LIDC (Armato III et al., 2011) datasets.

