# OpenReview forum: "Disentangling Shared Representations Improves Implicit Neural Representations for Medical Imaging"
_ICML.cc/2026/Conference — Submitted to ICML 2026_

### Official Review · Reviewer_hsHT · 2026-03-02

**Soundness:** 2
**Presentation:** 3
**Significance:** 4
**Originality:** 3
**Overall Recommendation:** 4
**Confidence:** 4

**Summary:**

This paper proposes an INR framework DisINR for medical imaging inverse problems, addressing the inefficiency of optimizing INRs and catastrophic forgetting issue. DisINR mainly introduces a shared encoder-decoder pair and subject-specific encoders. The shared pair is pre-trained to capture population priors. During test-time adaptation, the shared network is frozen, and only the new subject-specific encoder is optimized to reconstruct the target image.

**Compliance With Llm Reviewing Policy:**

Affirmed.

**Final Justification:**

The revisions and new experiments in Rebuttal have strengthened the manuscript. I am raising my score to Weak Accept, with the strong recommendation that the authors incorporate all the rebuttal materials and a deepened theoretical motivation into the final camera-ready version.

**Key Questions For Authors:**

Please see weaknesses, especially #2, #4, #5. I am open to raising my score if the authors can address these concerns.

**Limitations:**

Limitation in Appendix A.1 is insufficient for an ICML paper. In my view, the limitations of the proposed methods are:
1. Completely freezing the shared encoder and decoder during test-time adaptation may restricts the hypothesis space;
2. How to deal with the cases when encountering structures that are completely different from pre-training population priors.

**Strengths And Weaknesses:**

### **Strengths:**
1. The test-time adaptation strategy provides an elegant and fundamental structural solution to the catastrophic forgetting problem.
2. The factorized representation paradigm is flexible and effectively integrates with high-capacity backbones like NGP, which allows for high-fidelity local detail reconstruction.


### **Weaknesses:**
1. The claim in Abstract and Introduction that the framework "explicitly disentangles" shared and subject-specific representations. While according to Eq.4, DisINR only merges the features from the two encoders simply using concatenation. Intuitively, explicitly disentangling “shared” and “specific” usually needs mathematical constraints (e.g., mutual information minimization, orthogonality loss). Claiming explicit disentanglement based solely on concatenation and a freezing strategy is misleading. The authors should either provide qualitative/quantitative experiments to prove this disentanglement, or revise this terminology to a more appropriate one.
2. The fundamental baseline “Naive Fine-tuning” is missing. To truly demonstrate the necessity and advantage of introducing subject-specific encoder, comparing with a standard framework is necessary.
3. The claimed “strong generalization to unseen domains" might be not sufficiently rigorous. For fastMRI-T2w vs. fastMRI-FLAIR, FLAIR is essentially a T2-weighted sequence. They share same macroscopic anatomical priors. For DeepLesion vs. LIDC, DeepLesion is a comprehensive dataset that includes chest, abdomen, and pelvis lesions. Whereas LIDC focuses only on chest lung nodules. Anatomically, LIDC is largely a subset of DeepLesion. Therefore, perhaps the authors need to revise this claim, or provide additional experiments on truly challenging OOD scenarios.
4. This paper focus more on architectural design and performance. As a submission to ICML, providing motivation, intuition, rationale, or a comprehensive analytical dissection is necessary (why is this design rather than others? and why it works?).
5. In Table 2 fastMRI-T2w (AF=6x), DisINR achieves 48.42 dB PSNR, outperforming the second best method (40.22 dB)  8+ dB. Achieving such a leap in low-level vision performance only by altering feature concatenation and initialization strategies is a little counter-intuitive. It’s necessary to provide powerful explanation to explain cases like this.

---

> ### Author Rebuttal · Authors · 2026-03-30
>
> ### Thanks for your efforts and valuable comments. Below, we provide point-to-point responses to address your concerns.
>
> ---
> ## **Q1. Please clarify or provide evidence for the claimed "explicit disentanglement" of shared and subject-specific representations, or revise the terminology**
> Thanks for your insightful comments. We address this concern from three perspectives:
> - **Qualitative visualization of learned features:** We conduct additional experiments on the fastMRI-T2w dataset to inspect the representations learned during the pre-training stage. As shown in **Fig. R2 (https://anonymous.4open.science/r/DisINR)**, we visualize the shared and individual features using PAC. Specifically, the shared features capture smooth, population-level information, while the individual features encode subject-specific details. From the visualizations, it is evident that the shared and individual features are substantially different, which provides partial support for our disentanglement claim.
> - **Mathematical constraints:** The reviewer’s suggestion to incorporate mutual information minimization or orthogonality losses is insightful. Such explicit constraints could enforce stronger disentanglement, which we consider a promising direction for future research.
> - **Terminology adjustment:** As the current method lacks explicit mathematical constraints, we are open to revising the wording in the manuscript. For example, we will replace “explicit disentanglement” with “factorized representations” to better describe our approach.
>
> ## **Q2. "Naive Fine-tuning" is missing**
> To evaluate the necessity of the subject-specific encoder, we compare DisINR with the original NGP (naive INR without pretraining) and a fine-tuning variant of NGP (naive fine-tuning with pretraining). The latter is first pretrained on DeepLesion and then fully fine-tuned on test samples, similar to DisINR.
>
> Table R9 shows quantitative results. We observe that both versions of NGP achieve very similar performance, and both are about 2 dB lower in PSNR than DisINR. We also provide qualitative results in **Fig. R6 (https://anonymous.4open.science/r/DisINR)**. We will include these results in the revised paper.
>
> |Method|60 Views|90 Views|
> |---|---|---|
> |original NGP (naive INR, w/o pre-training)|37.30±1.20/0.960±0.009|38.51±1.08/0.971±0.006|
> |NGP (naive fine-tuning, w/ pre-training)|37.30±1.40/0.958±0.010|38.54±1.20/0.969±0.006|
> |DisINR (Ours)|**39.63±1.39/0.977±0.006**|**40.73±1.37/0.983±0.004**|
>
> *Table R9: Quantitative results (PSNR/SSIM) of original NGP, NGP with fine-tuning, and DisINR for sparse-view CT on the DeepLesion dataset.*
>
> ## **Q3. Validation on truly challenging OOD scenarios**
> We evaluate DisINR on a public COVID-19 CT dataset to test performance under extreme OOD scenarios with unseen structural anomalies (`see Q1 of Review 697j`).
>
> ## **Q4. Provide a stronger motivation to explain why this architectural design is chosen**
> The main motivation of our framework stems from two observations:
>
> - Classical INR methods train from scratch for each case and cannot embed population-level priors, limiting reconstruction quality;
> - Existing initialization-based methods suffer from severe catastrophic forgetting during fine-tuning, resulting in limited improvements.
>
> To address these issues, we learn shared and subject-specific representations via separate encoders. The shared encoder–decoder captures population-level priors, while freezing the shared components during test-time adaptation avoids catastrophic forgetting. Empirical results across multiple tasks and datasets confirm the effectiveness of this design.
>
> ## **Q5. Provide a stronger explanation for the large performance gain by DisINR for undersampled MRI on fastMRI-T2w dataset in Table 2**
> We provide two key points to clarify the large PSNR gain in Table 2:
>
> - **Cartesian sampling pattern:** Our experiments use a uniformly Cartesian pattern, which is clinically common but severely undersamples low-frequency components, resulting in strong aliasing artifacts. DisINR effectively learns low-frequency, population-level priors via the shared encoder-decoder, which constrains the solution space. Freezing these shared components further allows the priors to transfer to individual reconstructions, yielding substantial performance gains. This is also supported by qualitative results in Fig. 7, where aliasing artifacts are strongly reduced in DisINR reconstructions compared to baselines.
> - **Other sampling patterns:** For patterns that preserve more low-frequency information, such as radial sampling (`see Q4 of Review NDnx`), the improvements over STRAINER (second-best method) are smaller (+2~3 dB), as low-frequency loss is less severe.
>
> Overall, we attribute the large performance gain to the effective population-level priors learned by DisINR and the test-time freezing strategy, which together improve individual reconstructions, particularly when low-frequency components are severely undersampled.

---

> > ### Author Rebuttal · Reviewer_hsHT · 2026-04-03
> >
> > I would like to thank the authors for their extensive and constructive rebuttal. I appreciate the significant effort put into addressing the concerns raised in the initial review. The proposed method is undoubtedly effective and highly practical for medical imaging inverse problems. Overall, the authors have addressed almost all of the major concerns (Q1-Q3, Q5).
> >
> > ## Remaining Constructive Feedback for Q4
> > While the empirical validations are sufficient, I believe the response to Q4 still requires further elevation. The current motivation provided in the rebuttal largely restates what the architecture does (learning priors and freezing to avoid forgetting) and what the empirical outcomes are. However, it remains somewhat shallow.
> >
> > As an ICML submission, the manuscript would be significantly strengthened by delving into the deeper underlying mechanisms of why this specific design is fundamentally sound. I encourage the authors to expand their analysis, from the perspective such as structural priors, or how the frozen capacity restricts the hypothesis space to prevent feature collapse. Providing this deeper, more principled analysis rather than just empirical observations will greatly enhance the theoretical depth of the paper.

---

> > > ### Author Response · Authors · 2026-04-04
> > >
> > > We are glad to hear that most of your concerns have been addressed, and we thank the reviewer for the increased rating. We also thank the reviewer for the thoughtful suggestion to provide a deeper explanation of the underlying mechanism. Beyond the empirical observations, we believe the effectiveness of DisINR can be intuitively understood from two perspectives.
> > >
> > > - **Hypothesis space restriction via frozen priors.** By freezing the shared encoder–decoder during test-time adaptation, the optimization is effectively constrained to a subspace defined by the population-level priors learned during pretraining. This restriction limits the solution space of the reconstruction problem and helps avoid unstable solutions or feature collapse when fitting undersampled measurements.
> > > - **Structural prior decomposition.** DisINR separates population-level and subject-specific representations. The shared encoder captures common structural patterns across the cohort, while the subject-specific encoder adapts to individual variations, as shown in Fig. R2 (https://anonymous.4open.science/r/DisINR). This factorization allows the model to leverage strong structural priors while maintaining flexibility for patient-specific details.
> > >
> > > We agree with you that discussing these mechanisms more explicitly can further strengthen the motivation of the design, and we will incorporate these discussions into the revised manuscript.

---

### Official Review · Reviewer_pgZm · 2026-03-10

**Soundness:** 3
**Presentation:** 3
**Significance:** 3
**Originality:** 3
**Overall Recommendation:** 4
**Confidence:** 4

**Summary:**

DisINR introduces an interesting and novel, cohort-based learning paradigm for learning INR priors for inverse problems in medical imaging. Compared to the established transfer-learning / meta-learning frameworks in INRs, DisINR freezes a pre-trained encoder-/decoder to keep (!) the prior, and uses a second learnable "subject-specific" encoder to combine the learned prior when optimizing new instances during test time. The authors compare their proposed architecture to established frameworks on three different tasks, i.e. one 3D volume overfitting, and two measurement-based inverse problems for k-space / CT reconstruction, where they significantly outperform state-of-the-art.

**Compliance With Llm Reviewing Policy:**

Affirmed.

**Final Justification:**

The rebuttal has addressed my concerns adequately. I still remain positive about the paper and would like to keep my positive score.

**Key Questions For Authors:**

*Formulate 3–5 important questions whose answers would likely change your evaluation, clarify a confusing point, or address a critical limitation. Number each question.*
1. The authors claim that DisINR is architecture-agnostic (l.192) - but only show results for InstantNGP. Since InstantNGP is not particularly known for smooth interpolation, have the authors tried SIRENs or FFNs as well? A small test would be very appreciated. However, I acknowledge the NeRF ablation, which is a step in the right direction.
2. Regarding DisINR: One of the author's core hypothesis is that learned INR prior suffers from catastropic forgetting, when weights are updated during optimization. Sorry if I missed this, but did you ablate this for DisINR?
I think it would be interesting and relevant to see whether the performance gain actually comes from freezing the weights, or the new proposed archietctural improvement? Could you please ablate this?
3. Regarding STRAINER: Did you freeze shared weights for STRAINER as well? If not, could you give this an ablation as well?
4. In Ablation Fig 9. the comparison to Meta is omitted. Can the authors add this?
5. Since the authors do not use a SIREN based backbone for STRAINER / LearnedInit (Tancik), did they do an lr sweep?

**Limitations:**

Yes

**Strengths And Weaknesses:**

Strengths:
- The authors propose a simple, yet well-functioning framework, which consistenly yields highest performance across different tasks. I believe this constitutes a significant finding for the INR community, and may be transferable to other tasks, where learning priors is necessary or applicable.
- I like that the framework does not introduce complex optimization schemes, and doesn't significantly expand the model footprint, yet attains the highest performance scores.
- The presentation is relatively clear with minor exceptions. I would like to emphasize the inclusion of pseudo-code, which nicely complements the figures and makes the approach clearer. With few expections, the manuscipt is well written and comprehensive, particularly in articulating its findings, experiments and results. I appreciate the ablation regarding pretraining dataset size.

Weaknesses:
(major:)
- The authors tackle two problems (1) dependence on high quality images (l.064) and catastrophic forgetting (l.066). Regaridng the first, I would like to state that DisINR doesn't solve this differently than STRAINER; but may be better at modeling the prior itself. Regarding (2), authors claim that population priors are easily overwritten, if weights are optimized during test time. While I believe this, this is not ablated in the paper. This is surprising, since these experiments would not have been much work. E.g. train Strainer and DisINR once in frozen-/non-frozen settings. This would have allowed to quantify whether the improvements are induced by keeping the prior via weight freezing, or by incorporating a better suited architecture.
- While, technically, the authors compare against relevant baselines, they are omitting an important adjacent INR field that leverages cohort-based learning and learns priors. A lot of early, and concurrent works, learn cohort priors by using a conditioned auto-decoder archiecture [1-5], and thus embed the prior in the conditioned MLP [1,2], which is frozen at test time. Not only, do the authors not discuss this in related work, they also do not compare against (one of these) established frameworks, e.g. [1,2]. This is especially relevant, given that meta-learned approaches and conditioned approaches may be combined [1,2]. For instance, [6], a recent work from the authors of STRAINER, also compares against [4] in their ablation, featuring both meta-learned INRs and conditioned INR approaches.

(minor):
- Presentation: While the pseudo-code is extremely helpful in understanding the approach, I believe the figures could be clearer. For instance, Fig 1. / Fig. 2 do not properly convey how the architecture looks like (i.e. the concatenation of the encoder features as described in pseudocode). Both show classical encoder-/decoder approaches indicating a bottleneck. In INRs, where FFNs/SIREN architetcures are common this is a bit confusing, and could be improved. Given that the authors claim that their approach is agnostic to architecture, I would not (necessarily) show encoder-/decoder architectures (probably influenced by the choice of InstantNGP).
- In my opinion, the abstract is a bit clumsy and strays away from the main points. To me, the main points are: (1) Classical INRs do not incorporate learned proirs from datasets, as they are trained **per instance**. (2) Using meta-learning (or init), existing approaches learn cohort priors, but do not leverage it sufficiently, as they update (all )?! weights at test-time optimization. We introduce DisINR [...] which improves by (1) using pretrained frozen weights and by (2) incorporating a concatenation of cohort-/subject-specific features.
- I believe InstantNGP as a base architecture may not be the most optimal architecture for Meta-Learning / STRAINER. Why did the authors opt for it? InstantNGP is extremely good at overfitting signals, but it is not known to be great at interpolation tasks. It would have been interesting to see if this also holds across SIREN.
- Typos in l. 272 (an SOTA), l.314 (algorithm)

[1] Sitzmann, Vincent, et al. "Metasdf: Meta-learning signed distance functions." Advances in Neural Information Processing Systems 33 (2020): 10136-10147.

[2] Dupont, Emilien, et al. "From data to functa: Your data point is a function and you can treat it like one." arXiv preprint arXiv:2201.12204 (2022).

[3] Dupont, Emilien, et al. "Coin++: Neural compression across modalities." arXiv preprint arXiv:2201.12904 (2022).

[4] Stolt-Ansó, Nil, et al. "Nisf: Neural implicit segmentation functions." International Conference on Medical Image Computing and Computer-Assisted Intervention. Cham: Springer Nature Switzerland, 2023.

[5] Friedrich, Paul, Florentin Bieder, and Phlippe C. Cattin. "Medfuncta: Modality-agnostic representations based on efficient neural fields." arXiv e-prints (2025): arXiv-2502.

[6] Vyas, Kushal, Ashok Veeraraghavan, and Guha Balakrishnan. "Fit Pixels, Get Labels: Meta-learned Implicit Networks for Image Segmentation." International Conference on Medical Image Computing and Computer-Assisted Intervention. Cham: Springer Nature Switzerland, 2025.

---

> ### Author Rebuttal · Authors · 2026-03-30
>
> ### Thank the reviewer for the valuable comments. We are encouraged by your recognition of our work. Below, we provide point-to-point responses to address your concerns.
>
> ---
> ## **Q1. Validation on Different INR architectures**
> We evaluate the robustness of our DisINR framework across different INR architectures (`see Q2 of Review NDnx`).
> ## **Q2. Ablation study on weight freezing in DisINR**
> To study the effect of weight freezing on model performance, we additionally perform an ablation study on the DeepLesion dataset. As shown in Table R7, we make three observations:
>
> - Freezing the shared components in DisINR slightly improves performance (+1.47 dB in PSNR).
> - Regardless of freezing, DisINR consistently outperforms existing SOTA INR-based methods (NGP and STRAINER).
> - Freezing the shared components in STRAINER significantly reduces performance (about −8 dB in PSNR).
>
> We attribute these observations to the following reasons:
>
> - Without freezing the shared encoder-decoder, DisINR partially suffers from catastrophic forgetting, which reduces its performance.
> - By factorizing population-level and subject-specific representations using separate encoders, DisINR learns effective semantic representations, enabling improved reconstructions even without freezing the shared components.
> - STRAINER's single-encoder design cannot effectively separate shared and subject-specific representations, leading to ineffective transfer. Thus, STRAINER requires full fine-tuning, which is consistent with its original paper.
>
> The qualitative results are shown in **Fig. R4 (https://anonymous.4open.science/r/DisINR)**, confirming the above observations. We will include this ablation study in the revised paper.
>
> |Method|Frozen components|60 Views|
> |---|---|---|
> |NGP (naive INR)|N/A|37.30±1.20/0.960±0.009|
> |STRAINER (w/o Frozen)|N/A|36.65±1.24/0.950±0.012|
> |STRAINER (w/ Frozen)|Shared Encoder|28.40±2.34/0.832±0.057|
> |DisINR (w/o Frozen)|N/A|38.18±1.12/0.967±0.007|
> |DisINR (w/ Frozen)|Shared Encoder-Decoder|**39.63±1.39/0.977±0.006**|
>
> *Table R7: Quantitative results (PSNR/SSIM) of two baselines and DisINR ablating frozen components for sparse-view CT on the DeepLesion dataset.*
>
> ## **Q3. Difference between DisINR and STRAINER for dependence on high-quality images**
> Unlike STRAINER, which requires high-quality images for pretraining, DisINR can be pretrained using only a set of undersampled measurements (Note that recovering high-quality images from these undersampled measurements is non-trivial). As described in Sec. 3.2, DisINR incorporates a differentiable forward model to directly pretrain the shared encoder-decoder pair from the undersampled measurements. Hence, our framework effectively eliminates the need for high-quality images.
> ## **Q4. Please include a comparison to Meta in Ablation Fig. 9**
> We include the performance of Meta as the number of pretraining samples increases in Table R8. We observe that Meta's performance remains nearly unchanged with more pretraining samples. We attribute this to the relatively small pretraining size used in our experiments (N ≤ 50), whereas meta-learning-based methods like Meta typically require a much larger number of prior samples to achieve significant improvements. In contrast, DisINR consistently achieves stable performance gains thanks to its effective disentangled representations. We will include these results in the revised paper.
>
> |Method|N=5|N=10|N=50|
> |---|---|---|---|
> |Meta|40.43|40.40|40.49|
> |DisINR(Ours)|**47.92**|**48.76**|**50.41**|
>
> *Table R8: Performances (PSNR) of Meta and DisINR under different numbers of pre-training samples N for undersampled MRI with a Cartesian pattern of AF=6× on the fastMRI-T2w dataset.*
>
> ## **Q5. Missing discussion of cohort-based INR methods that learn dataset priors**
> We thank the reviewer for highlighting prior works on cohort-based or conditioned INRs. We will discuss them in detail in the revised paper to acknowledge their contributions. Our work focuses on medical image reconstruction, whereas many of these methods aim to learn general representations for tasks such as segmentation. Therefore, we did not include them as baselines. **In the medical imaging context, STRAINER remains the state-of-the-art INR method using population-level priors**, and we adopt it as the main baseline to demonstrate the advantages of DisINR.
> ## **Q6. Learning rate sweep for STRAINER and Meta**
> In our experiments, we carefully conduct hyperparameter searches for all baselines on the pre-training datasets to ensure a fair comparison. **To improve reproducibility, we will publicly release the source code**.
> ## **Q7. Clarification of contributions and motivation in Abstract**
> We thank the reviewer for the suggestion. We will revise the abstract to better highlight the limitations of classical INRs and meta-learning approaches, and how DisINR addresses them via frozen pretrained weights and disentangled cohort-/subject-specific representations.

---

> > ### Author Rebuttal · Reviewer_pgZm · 2026-04-02
> >
> > Thank you for the answers to my questions. My views regarding the strengths and weaknesses of the proposed method remain unchanged.

---

> > > ### Author Response · Authors · 2026-04-03
> > >
> > > Thank you for your response, and we appreciate that your assessment of the strengths and weaknesses remains unchanged. We are glad to see that your concerns are now marked as fully resolved. We hope this may be taken into consideration in the overall evaluation.

---

### Official Review · Reviewer_697J · 2026-03-12

**Soundness:** 3
**Presentation:** 4
**Significance:** 3
**Originality:** 3
**Overall Recommendation:** 4
**Confidence:** 3

**Summary:**

This paper proposes DisINR, a framework for improving implicit neural representations (INRs) with respect to medical inverse problems. Typically INRs suffer from two key challenges: 1) they depend on high quality imaging which is difficult to obtain and 2) experience catastrophic forgetting in that case-specific finetuning may overwrite general population-level knowledge. DisINR addresses this by disentangling how general and subject-specific representations are learned, using separate encoders to learn population- and subject-level knowledge with a shared decoder. During test time the shared encoder-decoder pair are frozen, while the subject-level encoder is optimized for a specific sample. Not to mention, DisINR works as a plug-and-play framework, as it is versatile with different INR backbones. DisINR is evaluated on three diverse medical imaging tasks, 3D volume fitting, under sampled MRI and sparse-view CT, demonstrating that it is a step forward in INR research.

**Compliance With Llm Reviewing Policy:**

Affirmed.

**Final Justification:**

I will keep my score as a weak accept because, while the rebuttal addressed several concerns regarding scalability and robustness, there is still a lack of targeted analysis into how well specific anomalies are reconstructed. Although the framework demonstrates strong soundness and originality through its disentangled architecture, further evidence is needed to ensure unique pathologies are not over-smoothed into population priors. Overall, the work remains a significant contribution to the INR field, but these remaining gaps limit an increase in score.

**Key Questions For Authors:**

1.	Have you evaluated the model’s ability to reconstruct rare pathologies or other structural anomalies, especially those absent from the pretraining set? It’s possible that the frozen shared encoder-decoder may be constrained in terms of the population priors it learns and “over-smooths” previously unseen anomalies which would be an important limitation to address for improved clinical utility.
2.	How does pretraining time scale as pretraining subject size increases? You demonstrate that DisINR performance scales with pretraining size, it would be good to demonstrate change in pretraining time complexity as well to support claims of scalability.
3.	In practice, the system matrix A probably varies between systems/machines used to collect data. Is it possible to evaluate the degradation in reconstruction quality if a different forward model is used between pretraining and test-time? Evaluating the sensitivity of the model to the system matrix would be helpful to understand robustness.

**Limitations:**

PSNR and SSIM are not holistic metrics when evaluating clinical utility, the authors should discuss the need for a radiologist observer study or more intentional clinical validation of reconstructed images.

**Strengths And Weaknesses:**

Strengths:
- Introduces an intentional architectural design of separate encoders to learn population priors and subject details. The adaptation during test-time leverages this design so that the general anatomical knowledge is appropriately integrated and catastrophic forgetting is addressed.
-	Able to learn without a large set of high quality images for pretraining. By using a differentiable forward model the authors include a sound application of physics-informed learning.
-	Authors demonstrate that DisINR is architecture-agnostic allowing different INR backbones to be integrated. They demonstrate improved performance with the NGP and NERF backbones.
-	Effectively combines existing methodologies such as INRs and differentiable forward models to demonstrate a novel insight on how to disentangle representations for medical imaging and improve reconstruction of high quality images from undersampled data.

Weaknesses:
- The paper does not fully discuss the limitation of pretraining time. It would be important to know how this training time scales with respect to pretraining dataset size, especially if it scales beyond six volumes to hundreds or thousands of subjects.
-	The paper relies on PSNR and SSIM to measure reconstruction quality. Although these metrics are standard they have their downsides in that they may poorly correlate with human perception and can miss high-frequency details respectively. Without a radiologist reader study or a more comprehensive set of evaluation metrics, the claim of improved clinical utility is hard to make.
-	The representation learned by the shared encoder-decoder is critical. There is potential risk that the model forces unique pathology into the representation space of average normal anatomy.

---

> ### Author Rebuttal · Authors · 2026-03-30
>
> ### Thank you for taking the time to review our work. We are pleased to receive your positive feedback. Below, we provide point-to-point responses to address your concerns.
>
> ---
> ## **Q1. Validation on unseen structural anomalies.**
> We use a public COVID-19 CT dataset to evaluate model performance on unseen structural anomalies. The dataset contains lung CT images from patients with COVID-19. We directly test five baselines and our DisINR, all pre-trained on DeepLesion dataset, on this unseen dataset.
>
> As shown in Table R3, DisINR achieves the best performance, significantly outperforming STRAINER, the second-best method, by +2 dB in PSNR. We provide qualitative results in **Fig. R5 (https://anonymous.4open.science/r/DisINR)**, where DisINR produces clearer images compared to baselines. Overall, the evaluation confirms the robustness of our framework on the structural anomalies. We will include these results in the revised paper.
>
> |Method|60 Views|90 Views|
> |------|-------|-------|
> |FBP|18.95±0.90/0.280±0.011|21.87±0.90/0.358±0.017|
> |SAX_NeRF|32.09±1.39/0.926±0.003|33.02±1.36/0.940±0.003|
> |NGP|31.50±1.30/0.916±0.004|32.45±0.97/0.933±0.005|
> |MetaNGP|31.55±1.23/0.916±0.004|32.43±1.03/0.932±0.005|
> |STRAINER|30.68±1.27/0.899±0.006|31.45±1.11/0.916±0.003|
> |DisINR (Ours)|**32.73±1.43/0.935±0.005**|**33.72±1.18/0.948±0.005**|
>
> *Table R3: Quantitative results (PSNR/SSIM) of five baselines and DisINR for sparse-view CT on an unseen COVID19 dataset.*
>
> ## **Q2. Discussion on limitations of pretraining time**
> Fig. 9 of main paper shows DisINR’s performance improves as the number of pretraining subjects increases (N = 5, 10, 50). We agree that pretraining time complexity is important for scalability. At each pretraining step, we randomly sample M ≤ N subjects (M = 5), and the number of pretraining epochs is kept the same across all N. **Thus, overall pretraining time does not increase significantly, taking roughly 30 minutes on a single NVIDIA RTX 4070 Ti for all cases.**
>
> While we will explore larger-scale pretraining in future work, DisINR already achieves strong performance with a small pretraining size (N ≤ 50), demonstrating the efficiency of its disentangled representation in capturing both population-level and subject-specific features.
> ## **Q3. Robustness of DisINR to forward model mismatch**
> Our DisINR is built on the INR framework, which supports flexible incorporation of different forward models. To test its robustness, we simulate a new sparse-view CT acquisition geometry. Compared with the old geometry in Table 5 (#Detectors=500, DSC=300 mm, DCD=300 mm), the new geometry uses #Detectors=400, DSC=400 mm, DCD=400 mm. We then directly evaluate DisINR and all baselines on this unseen geometry without additional pretraining.
>
> As shown in Table R4, all methods achieve stable performance. More importantly, DisINR still achieves the best performance, validating its robustness to forward model variations.
>
> Moreover, **we also evaluate DisINR for undersampled MRI under different sampling patterns** (`see Q4 of Review NDnx`).
>
> |Method|Old geometry|New geometry|
> |------|------------|------------|
> |FBP|24.29±1.03/0.363±0.045|25.87±0.87/0.393±0.043|
> |SAX_NeRF|38.32±1.48/0.968±0.009|38.77±1.12/0.970±0.006|
> |NGP|37.30±1.20/0.960±0.009|38.05±1.16/0.964±0.007|
> |Meta|37.45±1.07/0.962±0.007|38.09±1.12/0.964±0.007|
> |STRAINER|36.65±1.24/0.950±0.012|37.33±1.28/0.955±0.010|
> |DisINR (Ours)|**39.63±1.39/0.977±0.006**|**40.68±1.24/0.981±0.005**|
>
> *Table R4: Quantitative results (PSNR/SSIM) of five baselines and DisINR for sparse-view CT on the DeeepLesion dataset with an old and new acquisition geometries.*
>
> ## **Q4. More realistic evaluation metrics**
> PSNR and SSIM are widely used quantitative metrics in medical imaging, so we adopt them as the main criteria. However, we agree with the reviewer that they may not fully measure clinical utility. Therefore, following recent works in medical reconstruction, we also report LPIPS, a perceptual metric based on deep learning that better reflects human visual similarity. As shown in Tables R5 and R6, DisINR still achieves the best performance in terms of LPIPS. We will include these results in the revised paper.
>
> |Method|60 Views|90 View|
> |---|---|---|
> |FBP|0.3185±0.0307|0.2393±0.0287|
> |SAX_NeRF|0.0196±0.0075|0.0133±0.0047|
> |NGP|0.0233±0.0066|0.0159±0.0051|
> |MetaNGP|0.0226±0.0068|0.0156±0.0047|
> |STRAINER|0.0271±0.0090|0.0207±0.0070|
> |DisINR(Ours)|**0.0108±0.0040**|**0.0071±0.0022**|
>
> *Table R5: Quantitative results (LPIPS) of five baselines and DisINR for sparse-view CT on the DeeepLesion dataset.*
>
> |Method|AF=6x|AF=8x|
> |---|---|---|
> |ZF|0.2267±0.0196|0.2500±0.0208|
> |IMJENSE|0.0135±0.0212|0.0675±0.0292|
> |NGP|0.0192±0.0224|0.0831±0.0285|
> |MetaNGP|0.0174±0.0214|0.0797±0.0288|
> |STRAINER|0.0078±0.0130|0.0436±0.0217|
> |DisINR (Ours)|**0.0023±0.0074**|**0.0151±0.0165**|
>
> *Table R6: Quantitative results (LPIPS) of five baselines and DisINR for undersampled MRI on the fastMRI-T2w dataset.*

---

> > ### Author Rebuttal · Reviewer_697J · 2026-04-03
> >
> > Most questions/concerns were addressed in the rebuttal. The paper would benefit from a targeted analysis into how well a given anomaly is specifically reconstructed, which would address the weakness that the “model forces unique pathology into the representation space of average normal anatomy.”

---

> > > ### Author Response · Authors · 2026-04-04
> > >
> > > We thank the reviewer for the constructive and insightful comments. We appreciate the suggestion to further analyze how anomalies are reconstructed. We respond from three aspects:
> > >
> > > - **Physics-informed adaptation.** During test-time adaptation, DisINR incorporates the forward model, enforcing data consistency with the measurements. This physics-informed constraint helps mitigate the risk of forcing unseen anomalies into the space of average anatomy and improves robustness to structural variations.
> > >
> > > - **Diversity of prior data.** Increasing the diversity of the pretraining dataset (e.g., including both healthy cases and various pathologies) can further improve the model's ability to represent uncommon structures.
> > >
> > > - **Empirical observation.** Our preliminary experiments on OOD datasets (e.g., COVID-19 CT) suggest that DisINR remains robust to unseen structural anomalies, indicating that the learned representations do not simply collapse to normal anatomy.
> > >
> > > We agree that a more targeted analysis on anomaly reconstruction would be valuable, and we will include a more detailed discussion in the revised paper and future work.

---

### Official Review · Reviewer_NDnx · 2026-03-18

**Soundness:** 3
**Presentation:** 3
**Significance:** 3
**Originality:** 3
**Overall Recommendation:** 4
**Confidence:** 4

**Summary:**

The paper proposes DisINR, a framework that disentangles population-level and subject-specific components within implicit neural representations to improve medical image reconstruction. It learns a shared prior across subjects and adapts to new data by optimizing only a lightweight subject-specific module, enabling faster and more stable convergence. Experiments show improved reconstruction quality and efficiency over existing INR-based methods.

**Compliance With Llm Reviewing Policy:**

Affirmed.

**Final Justification:**

I thank the authors for their thorough and well-structured rebuttal. My main concerns have been satisfactorily addressed. In particular, the clarification of DisINR’s distinction from prior INR-based approaches, along with the additional experiments on different backbones (including SIREN) and sampling strategies (e.g., radial), significantly strengthen the paper.

While some directions (e.g., downstream evaluation in weight space) remain for future work, they are not critical to the core contributions. Overall, the rebuttal increases my confidence in the novelty, robustness, and empirical validity of the proposed method, and I am therefore updating my score to weak accept.

**Key Questions For Authors:**

1. Several prior works already explore patient-specific priors and shared representations in INR-based MRI reconstruction. Can the authors more clearly articulate what fundamentally differentiates DisINR from these approaches?
2. How does the method perform with other backbones such as SIREN, which are used in methods like Meta and STRAINER?
3. Have you considered evaluating the learned parameter space for downstream tasks? For instance, after fitting INRs (NGP, Meta, STRAINER, DisINR) on cropped regions of the DeepLesion dataset (e.g., lesion vs. non-lesion), could a transformer operating on weight space (as in fit-a-nef, see Papa et al., CVPR'24) be used for classification?
4. Please describe the hardware used for constructing the INRs, including memory footprint and whether parallel training of multiple INRs is supported.
5. How would the method perform under radial sampling, which is more robust to motion artifacts than Cartesian sampling? Would you expect similar improvements in such settings?

**Limitations:**

yes

**Strengths And Weaknesses:**

**Strengths**
1. The paper proposes a clear disentanglement between population-level and subject-specific representations within INRs, which is a well-motivated architectural choice
2. Freezing the shared encoder–decoder while optimizing only a lightweight subject-specific encoder is computationally appealing and leads to faster convergence
3. The method demonstrates consistent improvements in reconstruction quality (PSNR/SSIM) and convergence speed across multiple tasks and datasets

**Weaknesses**
1. The idea of incorporating patient-specific priors and shared representations in INR-based MRI reconstruction has already been previously explored in works such as “Accelerated Patient-specific Non-Cartesian MRI Reconstruction using Implicit Neural Representations” (Xu et al., International Journal of Radiation Oncology, Biology, Physics) and “Universal mapping and patient-specific prior implicit neural representation for enhanced high-resolution MRI in MRI-guided radiotherapy” (Li et al., Med Phys., 2025).
2. Although the method is claimed to be backbone-agnostic, experiments are largely centered on NGP. Validation on other INR architectures (e.g., SIREN) is limited, which weakens the claim of general applicability.
3. The paper does not investigate whether the learned representations are meaningful or transferable, focusing solely on image quality metrics. This is a missed opportunity given the emphasis on disentangled representations.
4. Important implementation details such as GPU type, memory footprint, parallelization strategy, and INR network architecture are not clearly reported, making it difficult to assess scalability, reproducibility and utilization.
5. The evaluation is limited to specific sampling schemes (e.g., Cartesian MRI). The robustness of the method under alternative sampling strategies (e.g., radial sampling) is not explored
6. It is unclear whether the INR is consistently trained to predict image intensity values from spatial coordinates across all tasks, or whether this formulation changes for MRI reconstruction, where the measurements are in k-space. If the latter is considered, the paper should clarify how complex-valued quantities are handled within the INR framework

---

> ### Author Rebuttal · Authors · 2026-03-30
>
> ### We thank the reviewer for the valuable comments. Below, we provide detailed point-by-point responses.
>
> ---
> ## **Q1. Differences between DisINR and previous INR methods with patient-specific priors**
> We thank the reviewer for pointing out these prior works. While related, they differ technically and in focus from DisINR:
> - **Xu et al., 2025:** This work directly represents prior k-space signals using INRs and then fine-tunes the network to each subject, following a naive fine-tuning paradigm. In contrast, DisINR recovers image signals through a differentiable forward model and learns shared and individual representations. Our design fundamentally mitigates catastrophic forgetting and leads to improved performance.
> - **Li et al., 2025:** This work proposes a supervised INR method for MRI super-resolution as a post-processing task. DisINR, however, is self-supervised and addresses reconstruction as a pre-processing task, without requiring ground-truth images.
>
> These distinctions show, although prior works exploit patient-specific priors, DisINR introduces a fundamentally different framework designed for self-supervised medical imaging. We will also discuss these works in the revised paper to fairly acknowledge their contributions.
> ## **Q2. Validation on different INR architectures**
> To evaluate the robustness of DisINR across different INR architectures, we implement DisINR and STRAINER using two representative INR backbones (NeRF and SIREN). For a fair comparison, vanilla INR, STRAINER, and DisINR are configured with comparable numbers of learnable parameters under each backbone. We perform careful hyperparameter tuning for all methods. Table R1 reports quantitative results. We make two key observations:
> - Across three different backbones, DisINR consistently achieves the best performance.
> - Using the NGP backbone significantly improves the performance of all three methods due to its larger network capacity.
>
> We also provide qualitative results in **Fig. R1 (https://anonymous.4open.science/r/DisINR)**, which further confirm the generalization of DisINR across different INR architectures. We will include this study in the revised paper.
>
> |Backbone|Method|PSNR|#Param.|
> |--------|------|----|-------|
> |NeRF|vanilla INR|37.03±2.23|0.34M|
> ||STRAINER|36.78±1.61|0.34M|
> ||DisINR (Ours)|**40.16±1.96**|**0.31M**|
> |SIREN|vanilla INR|35.73±1.15|0.46M|
> ||STRAINER|35.76±2.54|**0.43M**|
> ||DisINR (Ours)|**38.26±1.04**|0.44M|
> |NGP|vanilla INR|38.73±5.56|**4.92M**|
> ||STRAINER|41.75±2.33|4.97M|
> ||DisINR (Ours)|**49.61±3.82**|4.94M|
>
> *Table R1: Quantitative results of vanilla INR, STRAINER, and DisINR implemented by different backbones for undersampled MRI with a Cartesian pattern of AF=6x on the fastMRI-T2w dataset.*
> ## **Q3. Transferability of learned representations**
> We further evaluate the transferability of learned disentangled representations (`see Q1 of Review hsHT`). Using shared and subject-specific features for downstream tasks, such as classification, is an interesting direction. Our current work focuses on medical imaging, and downstream transfer is beyond this study’s scope, but we will discuss it in the revised paper.
> ## **Q4. Validation on radial sampling**
> We conduct additional experiments to evaluate model performance under different sampling trajectories. Specifically, the models are pre-trained on the fastMRI-T2w dataset using Cartesian sampling, and we directly test them under radial sampling (AF=10x) and Poisson sampling (AF=20x) without extra pre-training.
>
> As shown in Table R2, DisINR achieves the best performance across all cases, significantly outperforming the second-best method (STRAINER) by +3.34 dB and +2.07 dB in PSNR. We also provide qualitative results in **Fig. R3 (https://anonymous.4open.science/r/DisINR)**.
>
> |Method|Radial (AF=10x)|Poisson (AF=20x)|
> |------|---------------|---------------|
> |ZF|18.84±1.95/0.520±0.056|16.57±1.53/0.402±0.043|
> |IMJENSE|34.51±3.42/0.968±0.010|34.78±3.92/0.951±0.014|
> |NGP|31.94±3.50/0.926±0.024|30.99±3.38/0.843±0.055|
> |MetaNGP|32.01±3.52/0.924±0.026|30.97±3.58/0.841±0.063|
> |STRAINER|33.88±3.20/0.960±0.011|34.88±3.71/0.955±0.015|
> |DisINR (Ours)|**37.22±2.79/0.976±0.006**|**36.95±3.88/0.964±0.011**|
>
> *Table R2: Quantitative results (PSNR/SSIM) of five baselines and DisINR for undersampled MRI on the fastMRI-T2w dataset.*
>
> ## **Q5. INR Implementation details and hardware setup**
> The INR implementation details are provided in Sec. A.4 of Appendix. All experiments are conducted on a single NVIDIA RTX 4070 Ti GPU (12 GB). Across all experiments, the RAM usage does not exceed 16 GB. We will provide these details in the revised paper. **Also, we will publicly release our code to improve reproducibility.**
>
> ## **Q6. INR for complex-valued MRI k-space data**
> MRI images are complex-valued, so DisINR learns a function mapping spatial coordinates to real and imaginary components, with the decoder having two output channels. This is standard in INR-based MRI reconstruction.

---

> > ### Author Rebuttal · Reviewer_NDnx · 2026-04-05
> >
> > I thank the authors for their clear and constructive rebuttal. They have adequately addressed my main concerns, and I find their clarifications and additional explanations satisfactory.

---

> > > ### Author Response · Authors · 2026-04-05
> > >
> > > Thank you for your positive feedback and for raising your score. We are glad to hear that our response has adequately addressed all your concerns. We truly appreciate your constructive comments, which have helped us improve the paper.

---

### Decision · Program_Chairs · 2026-04-30

**Decision:**

Reject

**Comment:**

While the reviewers agreed that the paper is technically solid, clearly written, and empirically strong on several medical imaging reconstruction tasks, they also raised serious concerns about the paper’s novelty and the strength of evidence supporting its main claims. I have read the rebuttal and taken it into account. The rebuttal strengthens the empirical case, but it does not fully resolve the central novelty concerns.

In particular, the broad methodological idea is too close to prior INR work on transferable/shared priors, subject-specific adaptation, and prior-informed reconstruction [1-4]. The strongest remaining novelty appears to be pre-training shared modules directly from raw measurements via a differentiable forward model, rather than relying on high-quality image supervision. However, this point is not directly isolated by a controlled ablation. As a result, the paper shows that the proposed method performs well overall, but does not convincingly demonstrate that its narrowest and most defensible novelty claim is the source of the gains.

Overall, I find the paper promising and technically competent. However, the concerns about novelty and claim validation remain substantial. Therefore, the paper cannot be accepted to the conference at this time. I encourage the authors to revise the framing of the contribution, more carefully position the work with respect to the closest INR literature, and directly ablate the role of measurement-domain pretraining in a future revision.

## Additional AC Assessment
My additional concern is that, in light of the closest prior work, most ingredients of the method already appear in related form: transferable/shared INR features in STRAINER [1], prior-informed medical INR reconstruction in NeRP [2] and PINER [3], self-supervised measurement-domain INR reconstruction in IMJENSE [4], and broader meta-initialization/adaptation ideas in [6,7]. In my view, the only clearly plausible remaining novelty is the specific combination of cohort-level measurement-domain pretraining, frozen shared modules, and subject-specific adaptation. But this exact point is not isolated experimentally, especially given related joint/shared-latent INR reconstruction work [5]. This leaves the narrowest novelty claim under-validated.

## References
[1] K. Vyas, A. I. Humayun, A. Dashpute, R. G. Baraniuk, A. Veeraraghavan, and G. Balakrishnan, “Learning Transferable Features for Implicit Neural Representations,” NeurIPS, 2024.

[2] L. Shen, J. Pauly, and L. Xing, “NeRP: Implicit Neural Representation Learning with Prior Embedding for Sparsely Sampled Image Reconstruction,” IEEE Transactions on Neural Networks and Learning Systems, 2022.

[3] B. Song, L. Shen, and L. Xing, “PINER: Prior-Informed Implicit Neural Representation Learning for Test-Time Adaptation in Sparse-View CT Reconstruction,” WACV, 2023.

[4] R. Feng, Q. Wu, J. Feng, H. She, C. Liu, Y. Zhang, and H. Wei, “IMJENSE: Scan-specific Implicit Representation for Joint Coil Sensitivity and Image Estimation in Parallel MRI,” IEEE Transactions on Medical Imaging, 2023.

[5] J. Shi et al., “Implicit Neural Representations for Robust Joint Sparse-View CT Reconstruction,” Transactions on Machine Learning Research, 2024.

[6] M. Tancik et al., “Learned Initializations for Optimizing Coordinate-Based Neural Representations,” CVPR, 2021.

[7] V. Sitzmann et al., “MetaSDF: Meta-Learning Signed Distance Functions,” NeurIPS, 2020.